# Stem cell transplantation rescued a primary open-angle glaucoma mouse model

Siqi Xiong[1,2,3], Ajay Kumar[1], Shenghe Tian[1], Eman E Taher[1,4], Enzhi Yang[1], Paul R Kinchington[1], Xiaobo Xia[2,3], Yiqin Du[1,5,6]*

[1]Department of Ophthalmology, University of Pittsburgh, Pittsburgh, United States; [2]Eye Center of Xiangya Hospital, Central South University, Changsha, China; [3]Hunan Key Laboratory of Ophthalmology, Changsha, China; [4]Research Institute of Ophthalmology, Giza, Egypt; [5]Department of Developmental Biology, University of Pittsburgh, Pittsburgh, United States; [6]McGowan Institute for Regenerative Medicine, University of Pittsburgh, Pittsburgh, United States

**Abstract** Glaucoma is a leading cause of irreversible blindness. In this study, we investigated if transplanted stem cells are able to rescue a glaucoma mouse model with transgenic myocilin Y437H mutation and explored the possible mechanisms. Human trabecular meshwork stem cells (TMSCs) were intracamerally transplanted which reduced mouse intraocular pressure, increased outflow facility, protected the retinal ganglion cells and preserved their function. TMSC transplantation also significantly increased the TM cellularity, promoted myocilin secretion from TM cells into the aqueous humor to reduce endoplasmic reticulum stress, repaired the TM tissue with extracellular matrix modulation and ultrastructural restoration. Co-culturing TMSCs with myocilin mutant TM cells in vitro promoted TMSCs differentiating into phagocytic functional TM cells. RNA sequencing revealed that TMSCs had upregulated genes related to TM regeneration and neuroprotection. Our results uncovered therapeutic potential of TMSCs for curing glaucoma and elucidated possible mechanisms by which TMSCs achieve the treatment effect.

*For correspondence: duy@upmc.edu

## Introduction

Primary open-angle glaucoma (POAG), the most common type of glaucoma with a prevalence of 0.5–7.0% in adults, can result in damage of retinal ganglion cells (RGCs) and irreversible vision loss (*Broman et al., 2008*; *Quigley, 2006*). The progression of POAG has been demonstrated to be correlated with elevated intraocular pressure (IOP) (*Heijl et al., 2002*), which is associated with reduced trabecular meshwork (TM) cellularity (*Alvarado et al., 1984*; *Alvarado et al., 1981*), malfunction of TM phagocytosis (*Buller et al., 1990*) and abnormal deposition of extracellular matrix (ECM) (*Gong, 2016*; *Keller et al., 2009*). Replenishment of the TM cells with stem cells and restoration of the TM function offers a novel alternative approach to treat POAG (*Abu-Hassan et al., 2015*; *Du et al., 2012*; *Du et al., 2013*; *Kelley et al., 2009*; *Yun et al., 2018*; *Zhou et al., 2020*; *Zhu et al., 2016*).

Trabecular meshwork stem cells (TMSCs) have their special niche located at the anterior TM tissue beneath the Schwalbe's line (*Braunger et al., 2014*; *Raviola, 1982*; *Sundaresan et al., 2019*; *Yun et al., 2016*) and have been successfully isolated and characterized (*Castro and Du, 2019*; *Du et al., 2012*). TMSCs maintain stem cell characteristics and regenerative capacity after long-term cryopreservation (*Kumar et al., 2020*), which can be an effective source for cell-based therapy. After intracameral injection, TMSCs exhibit the preference to home to the TM region in wild-type mice (*Du et al., 2013*) and to laser-damaged TM tissue, which is correlated with CXCR4/SDF1 chemokine

axis (*Yun et al., 2018*). Moreover, TMSCs can improve the outflow facility in a mouse model with laser-induced TM damage (*Yun et al., 2018*). However, the mechanisms for TMSCs repairing the diseased TM and restoring TM function in POAG have not yet been resolved. Intriguingly, the pathogenesis of POAG is apparently different from that of laser-induced glaucoma (*Liesenborghs et al., 2020*). Hence, exploring the therapeutic effect of TMSCs on models of POAG and uncovering mechanisms underlying it are crucial steps for future clinical therapies for treating glaucoma.

Several factors, such as environment and genetics, have been found to contribute to the occurrence and development of POAG (*Janssen et al., 2013*). Mutations in the gene encoding myocilin (Myoc) have been confirmed to be associated with glaucoma (*Tamm, 2002*) which are responsible for 4% of adult-onset POAG and 10% of juvenile-onset POAG. Although Myoc mutation glaucoma is a subtype of POAG, the pathophysiology of POAG in common is associated with reduced TM cellularity, abnormal deposition of ECM and increased IOP. Myoc mutations alter the structure of Myoc protein and result in the retention of misfolded Myoc in the endoplasmic reticulum (ER) of TM cells. The accumulated protein can then induce ER stress in TM cells, which is related to glaucoma (*Peters et al., 2015*). ER stress can also lead to TM dysfunction, abnormal synthesis and turnover of ECM and loss of RGCs (*Fingert et al., 2002*). A mouse POAG model with transgenic-Myoc Y437H mutation (Tg-MyocY437H) closely mimics the pathophysiology of human Myoc-associated glaucoma (*Zhou et al., 2008*; *Zode et al., 2012*; *Zode et al., 2011*). Here, we report that human TMSCs that are intracamerally transplanted to the Tg-MyocY437H POAG mice, can repopulate the TM cells, repair the abnormal TM tissue and preserve the function of RGCs. By analyzing the RNA sequencing (RNAseq) data from three strains of human TMSCs and corneal fibroblasts from different donors, we have identified the expression differences for unveiling the TMSC regeneration mechanisms.

## Results

### TMSCs reduce IOP and increase outflow facility of the Tg-MyocY437H mice

Human TMSCs were isolated as previously described (*Yun et al., 2018*) and characterized by flow cytometry to confirm the positive expression of stem cell markers CD73, CD90, CD105, CD166, and negative expression of CD34 and CD45 as previously reported (*Kumar et al., 2020*; *Yun et al., 2018*). To investigate therapeutic effect of TMSCs on POAG, human TMSCs at passage three or four were injected into the anterior chamber of the Tg-MyocY437H mice (*Figure 1—figure supplement 1*) when they were at age of 4 months. Age-matched wild-type (WT) mice served as control. The baseline IOP of 4-month-old Tg-MyocY437H mice was 16.5 ± 0.44 mmHg (*Figure 1A*), which was significantly higher than that of WT mice (12.38 ± 0.41 mmHg, p<0.0001). Tg-MyocY437H mice transplanted with TMSCs started to lower IOP from 1 month after stem cell transplantation, and IOP decreased to 13.30 ± 0.42 mmHg (Tg-TMSC) at 1 month. This was close to that in WT mice (WT, 12.30 ± 0.45 mmHg, p=0.4576). The IOP was significantly lower than IOP of Tg-MyocY437H mice without treatment (Tg, 15.23 ± 0.64 mmHg, p=0.0254) and IOP of Tg-MyocY437H mice with medium only injection (Tg-Sham, 16.00 ± 0.38 mmHg, p=0.0005). There was no statistically significant difference of IOPs between untreated (Tg, 15.23 ± 0.64 mmHg) and sham (Tg-Sham, 16.00 ± 0.38 mmHg, p=0.6715). Indeed, the IOP of Tg-MyocY437H mice at 2 months after TMSC transplantation (Tg-TMSC, 12.65 ± 0.36 mmHg) reduced to the same level as that of WT (13.03 ± 0.39 mmHg, p=0.9425) and was lower than untreated Tg-MyocY437H (Tg, 14.96 ± 0.61 mmHg, p=0.0042) and sham injected mice (Tg-sham, 15.30 ± 0.21 mmHg, p=0.0006).

Meanwhile, we measured mouse night IOP which was more obviously elevated than day-time IOP as reported before (*Zode et al., 2011*). In consistent with the day-time IOP, there was a significant difference of the baseline night IOP between 4-month-old Tg-MyocY437H mice (*Figure 1B*, 17.73 ± 2.25 mmHg) and age-matched WT mice (13.67 ± 2.77 mmHg, p<0.0001). Two months after TMSC transplantation, the night IOP of Tg-MyocY437H mice (11.75 ± 2.83 mmHg) reduced to the same level as that of WT mice (11.55 ± 2.52 mmHg, p=0.9925).

To further elucidate if TMSCs reduced IOP in Tg-MyocY437H mice via regulating the conventional outflow pathway (the TM and the Schlemm's canal), we examined the outflow facility of the eyes from all groups. As shown in *Figure 1C* and Figure 1-figure supplement 2, Tg-MyocY437H mice

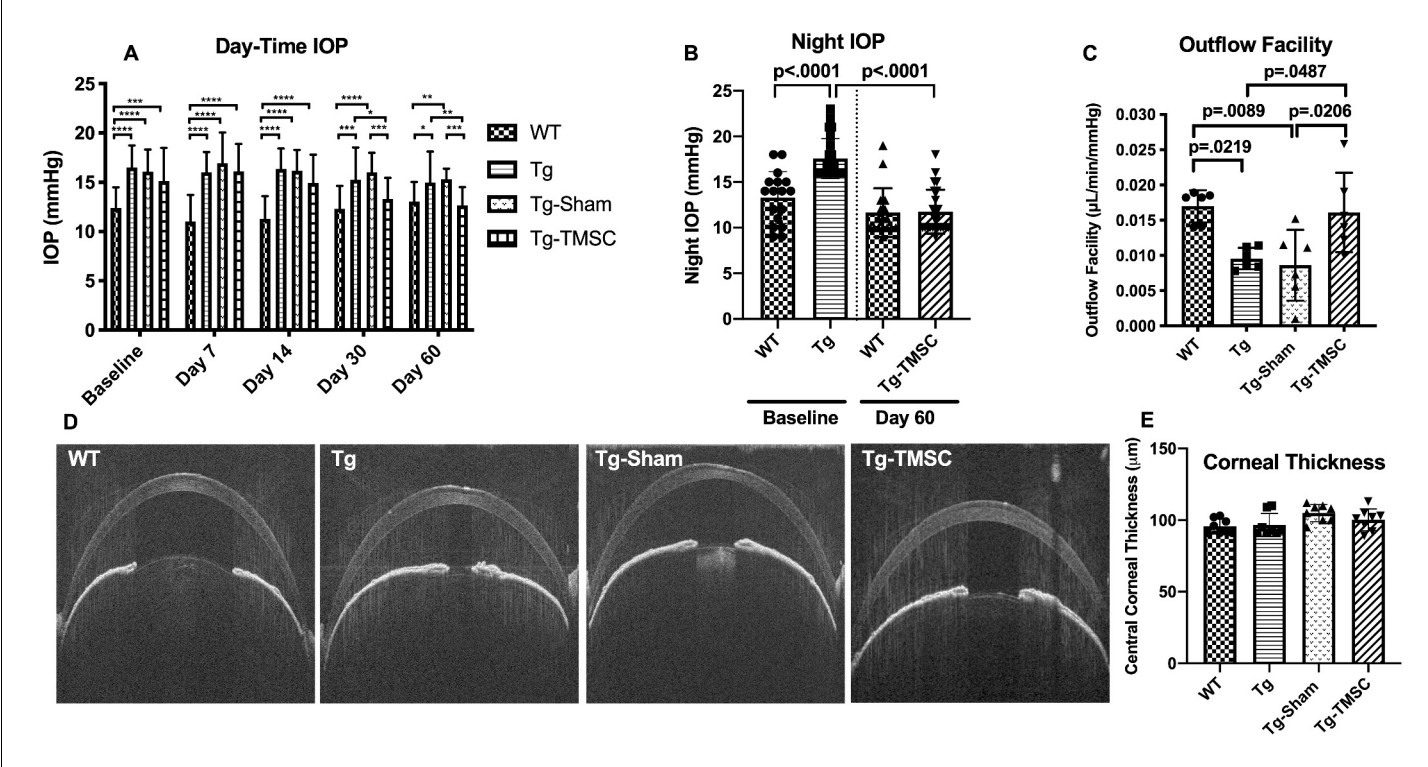

**Figure 1.** Transplanted TMSCs reduce the IOP and increase the outflow facility of Tg-MyocY437H mice. (A) Day-time IOP was measured in the wildtype mice (WT, n=26), Tg-MyocY437H mice (Tg, n=26), Tg mice treated with basal medium (Tg-Sham, n=26) and Tg mice with TMSC transplantation (Tg-TMSC, n=26). (B) Night IOP was measured in WT mice (n=17) and Tg-MyocY437H mice (n=24) before the treatment and 2 months post treatment. Data are presented as mean ± SD. (C) Outflow facility was evaluated at 2-month after TMSC transplantation (n=6 eyes/group). (D) Representative pictures of anterior OCT show the corneal thickness and anterior chamber angle in the mice at 2 months after transplantation. (E) The central corneal thickness was calculated from the OCT images (n=8 eyes/group). Data are presented as mean ± SD. Two-way ANOVA (A) or one-way ANOVA (B,C,E) followed by Tukey's multiple comparisons test. *p<0.05, **p<0.01, ***p<0.001, **** p<0.0001.

The online version of this article includes the following source data and figure supplement(s) for figure 1:

**Source data 1.** Raw day-time IOP data for *Figure 1A*; Raw night IOP data for *Figure 1B*; Individual outflow facility for *Figure 1C*; Individual central corneal thickness for *Figure 1E*.

**Figure supplement 1.** Genotyping of transgenic Myoc Y437H mice by polymerase chain reaction (PCR).

**Figure supplement 2.** Representative perfusion outflow data from each mouse group.

displayed higher outflow resistance with lower outflow facility (Tg, 0.010 ± 0.001 µL/min/mmHg), while the WT mice showed higher outflow facility (WT, 0.017 ± 0.001 µL/min/mmHg, p=0.0219). Tg-MyocY437H mice with TMSC transplantation for 2 months showed a significant facilitated outflow (Tg-TMSC, 0.016 ± 0.002 µL/min/mmHg) as compared to untreated (Tg, 0.010 ± 0.001 µL/min/mmHg, p=0.0487) and medium injected Tg-MyocY437H mice (Tg-Sham, 0.009 ± 0.005 µL/min/mmHg, p=0.029). This confirms that TMSCs reduced IOP via improvement of the TM-Schlemm's canal conventional outflow pathway.

Since corneal thickness can affect the accuracy of IOP measurement, anterior OCT was adopted to evaluate the thickness of the cornea, and this revealed that the corneas had the same thickness among all groups (*Figure 1D–E*). Anterior synechia, which is an important factor for assessing the efficacy and side effect of stem cell transplantation, was not found by OCT examination in the mice with intracameral injection of TMSCs or sham (*Figure 1D*).

## TMSCs prevent the RGC loss and preserve the RGC function of the Tg-MyocY437H mice

Preservation and rescuing of RGC function is the goal for the treatment of POAG, so evaluating the function of RGCs is critical in assessing the therapeutic effect of TMSCs on POAG. We used the

Celeris (Diagnosys LLC) to examine the pattern electroretinogram (PERG), an optimal approach to detect RGC function. In PERG by Celeris, the P1 amplitude represents the RGC function (*Figure 2A–B*). WT mice at 6 month of age had the P1 amplitude at 10.84 ± 0.86 μV in the PERG recording, while 39.4% of the RGC function was lost in 6-mongth old Tg-MyocY437H mice as calculated with the PERG (Tg, P1 = 6.56 ± 0.63 μV, p=0.0001). Two months after TMSC transplantation, nearly 90% of the RGC function of the Tg-MyocY437H mice was preserved (Tg-TMSC, P1 = 9.20 ± 0.45 μV; WT, P1 = 10.84 ± 0.86 μV, p=0.1934) (*Figure 2A–B*). Furthermore, we counted the RGC numbers on 5 μm paraffin sections (*Figure 2C*, *Figure 2—figure supplement 1*). There were 70.48 ± 2.26 RGC cells/mm in the retina of 6-month-old WT mice, and 49.22 ± 1.79 cells/mm in that of 6-month-old Tg-MyocY437H mice with RGC loss (p<0.0001). The RGCs were preserved/rescued by TMSC transplantation in Tg-MyocY437H mice with the RGC number increased to 60.60 ± 1.25 cells/mm (p<0.0001 as compared to untreated Tg-MyocY437H mice). This confirms that TMSC transplantation prevented/rescued the RGC loss and preserved the RGC function in the Tg-MyocY437H mice.

## TMSCs increase the TM cellularity of the Tg-MyocY437H mice

To further investigate the mechanisms by which TMSCs reduced IOP and restored outflow facility, we evaluated the TM cellular density in the mouse anterior segment sections. TM cellular density of the Tg-MyocY437H mice has been reported to be decreased due to apoptosis of TM cells in Tg-MyocY437H mice (*Zode et al., 2011*). Collagen IV was used as a marker to define the area of the TM tissue (*Zhu et al., 2016*). We combined the phase contrast black-white images with collagen IV staining to accurately identify the TM region and count the cells within the TM region on the paraffin sections. As shown in *Figure 3A–B*, the average number of the cells in the TM region was 35.33 ± 2.50 cells/section in 6-month-old WT mice (n = 10), while 42% and 51% of reduction of the TM cellularity was observed in the age-matched Tg-MyocY437H mice (Tg, 20.33 ± 1.11 cells/section, p<0.0001) and Tg-MyocY437H mice with Sham injection (Tg-Sham, 17.08 ± 1.18 cells/section,

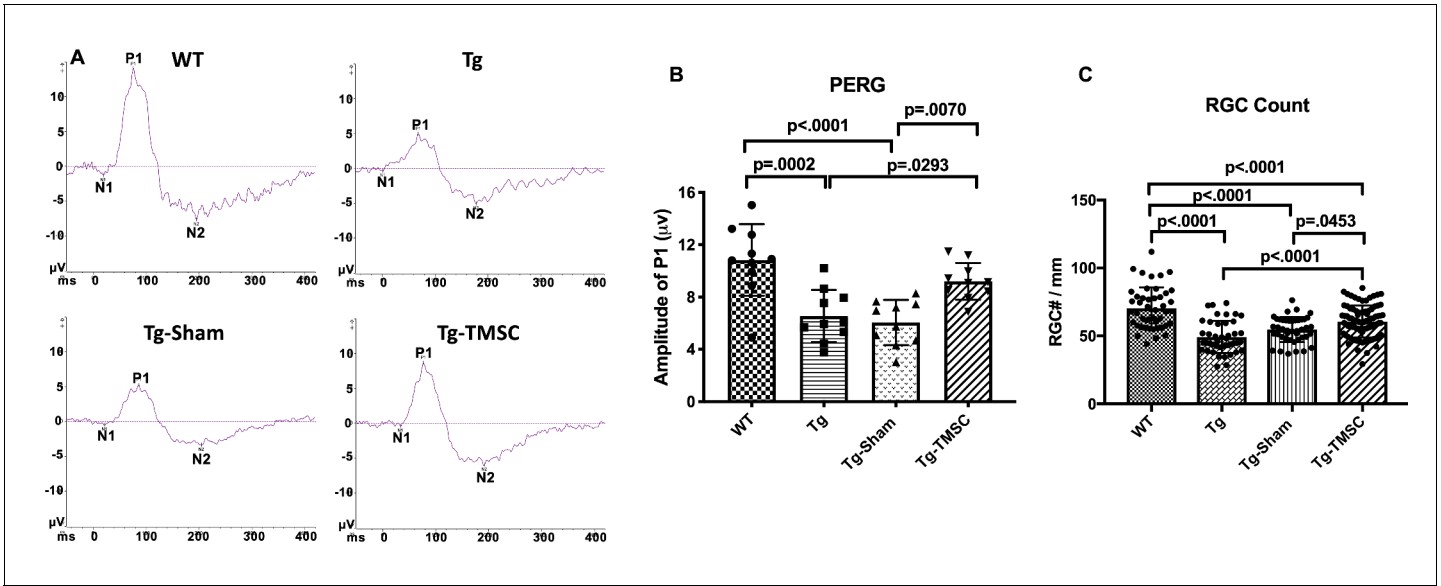

**Figure 2.** TMSCs preserve the RGC function and prevent RGC loss in Tg-MyocY437H mice. The function of RGCs in the mice was evaluated by pattern electroretinogram (PERG). (**A**) Representative examples of PERG from different groups at 2 months after transplantation. (**B**) Bar graphs of averaged P1 amplitude in PERG (n = 10 eyes/group). (**C**) RGC numbers counted on the retinal sections in each group (n = 12–16 sections/eyes, and 4–6 eyes/group). Data are presented as mean ± SD. One-way ANOVA followed by Tukey's multiple comparisons test. WT: wild-type mice, Tg: Tg-MyocY437H mice, Tg-Sham: Tg mice with medium injection, Tg-TMSC: Tg mice with TMSC injection.

The online version of this article includes the following source data and figure supplement(s) for figure 2:

**Source data 1.** Individual P1 amplitude in pattern electroretinogram for *Figure 2B*; Individual retinal ganglion cell count of each section for *Figure 2C*.
**Figure supplement 1.** Transplanted TMSCs rescue RGCs and prevent neurodegeneration in Tg-MyocY437H mice.

p<0.0001), respectively. TMSC transplantation on Tg-MyocY437H mice increased the TM cell number to 28.86 ± 1.46 cells/section (p=0.0028 vs Tg mice, p<0.0001 vs Tg-Sham).

## TMSCs differentiate into TM cells after transplantation in the Tg-MyocY437H mice

To examine if transplanted TMSCs differentiate into functional TM cells to contribute to the TM cellularity increase and TM function restoration, we labeled TMSCs with DiO and injected the cells into the anterior chamber of Tg-MyocY437H mice. Two months after TMSC transplantation, we detected DiO-labeled green cells at the mouse TM region (*Figure 4A–C*). Some of the DiO+ TMSCs expressed AQP1 and CHI3L1. Anti-AQP1 antibody detects both mouse and human AQP1 antigen while anti-CHI3L1 antibody used only detects human antigen (*Figure 4A*). Some of the TMSCs were positive to anti-Ki67 antibody staining (*Figure 4B*) indicating the transplanted TMSCs were active in proliferation 2 months after transplantation. TUNEL staining shows there were many TUNEL staining apoptotic cells in the Tg-MyocY437H mouse TM tissue while most of the injected TMSCs and mouse TM tissues were not stained with TUNEL (*Figure 4C–D*) indicating the injected TMSCs were alive and protective to the endogenous TM cells.

## TMSCs facilitate myocilin secretion and remodel the extracellular matrix (ECM) in the Tg-MYOCY437H mice

Tg-MyocY437H mouse model is characterized by the retention of misfolded Myoc protein in the ER of the TM cells and reduced secretion of Myoc protein into the aqueous humor (*Zode et al., 2011*), which are associated with the IOP elevation in this mouse model. Immunofluorescent staining on the paraffin sections showed that Myoc protein was not detectable in the TM region of WT mice, but it was obviously expressed in the TM region of the untreated and sham injected Tg-MyocY437H mice (*Figure 5A*). However, Myoc expression in the TM region was reduced in the Tg-MyocY437H mice receiving TMSC transplantation (*Figure 5A*). Western blotting on the limbal tissue confirmed the immunostaining results and showed that TMSC transplantation reduced Myoc protein accumulation in the limbus of the Tg-MyocY437H mice (*Figure 5B*). In contrast, very low levels of Myoc protein

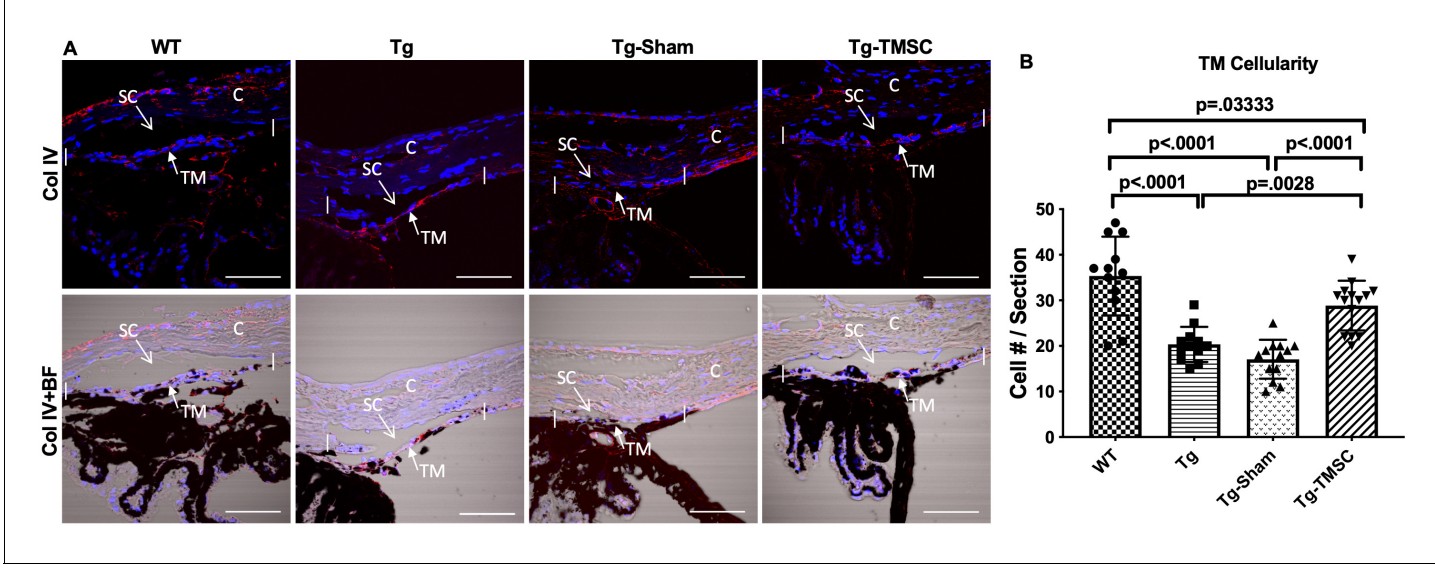

**Figure 3.** TMSCs increase the TM cellularity in the Tg-MyocY437H mice. (**A**) Evaluation of the cellular density in the mouse TM region. Sections of the anterior segment were immunostained with collagen IV (red) and DAPI (blue). The TM region was determined by bright-field (BF) image together with collagen IV staining in the region between the two white vertical lines. Scale Bars, 50 μm. (**B**) The TM cellularity was averaged (n=12-14/group) and displayed as the number of cells in the TM region per section. Data are presented as mean ± SD. One-way ANOVA followed by Tukey's multiple comparisons test. **C**: cornea, SC: Schlemm's canal, TM: trabecular meshwork.

The online version of this article includes the following source data for figure 3:

**Source data 1.** Individual TM cell count of each section for *Figure 3B*.

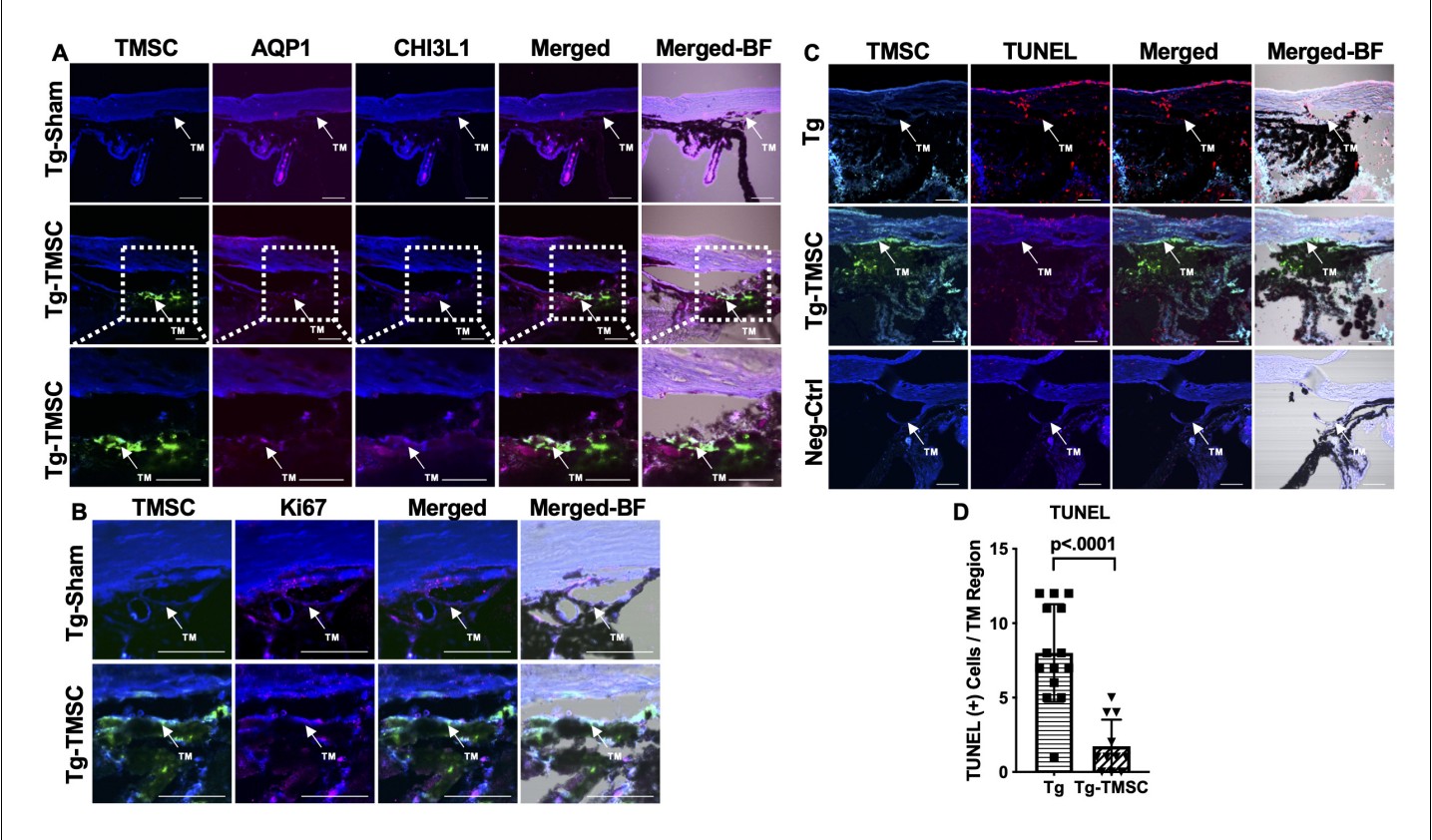

**Figure 4.** Transplanted TMSCs differentiate into TM cells and viable up to 2 months post-transplantation. (A) AQP1/CHI3L1 immunofluorescent staining shows integration of transplanted TMSCs (DiO+, green) into the TM and differentiation of TMSCs into TM cells with expression of AQP1 (red) and CHI3L1 (magenta). (B) Ki67 staining shows part of the transplanted TMSCs (green) positive to Ki67 (red) in the TM while few of the TM cells in the Tg-Sham were Ki67+ too. (C) TUNEL staining shows some of the corneal cells and TM cells in the Tg mice were positive to TUNEL (apoptosis) while the transplanted TMSCs (green) in the Tg-TMSC were viable as the TMSC population was TUNEL negative. Scale bars, 50 μm. (D) Quantification of TUNEL + cells in the TM region of both Tg-MYocY437H mice without treatment (Tg) and with TMSC transplantation (Tg-TMSC). Data are presented as mean ± SD. Student t-test. TM: trabecular meshwork.

The online version of this article includes the following source data for figure 4:

**Source data 1.** Individual TUNEL-positive cells per TM region per section for *Figure 4D*.

could be detected in the aqueous humor of the Tg-MyocY437H and Tg-Sham mice, but Myoc protein was increased dramatically in the aqueous humor of WT and TMSC transplanted Tg-MyocY437H mice (Tg-TMSCs; *Figure 5C*). This indicates that TMSC transplantation can enhance the secretion of Myoc protein from the TM into the aqueous humor in the Tg-MyocY437H mice.

Abnormal deposition of ECM in the TM tissue is known to contribute to IOP elevation. Indeed, increased levels of fibronectin and elastin were found in the limbus of the Tg-MyocY437H and Tg-Sham mice, while collagen IV remained at the similar levels as that in the WT mice (*Figure 5D*). However, TMSC transplantation downregulated the expression of fibronectin and elastin in the TMSC-treated Tg-MyocY437H mice to the levels in the WT mice (*Figure 5D*). This demonstrates that TMSCs could change the ECM components in the Tg-MyocY437H mice.

## The effect of TMSCs on ER stress in the TM of Tg-MyocY437H mice

To determine if TMSC transplantation could reduce ER stress in the TM of the Tg-MyocY437H mice, Western blotting was employed to detect the expression of ER stress markers in the mouse limbal tissue including the TM. The levels of CHOP and GRP78 were significantly increased in the limbal tissue of Tg-MyocY437H and Tg-Sham mice in comparison with the WT mice (*Figure 6A*). The expression of CHOP and GRP78 in the limbal tissue with TMSC injection was not significantly reduced as compared to the Tg-MyocY437H mice.

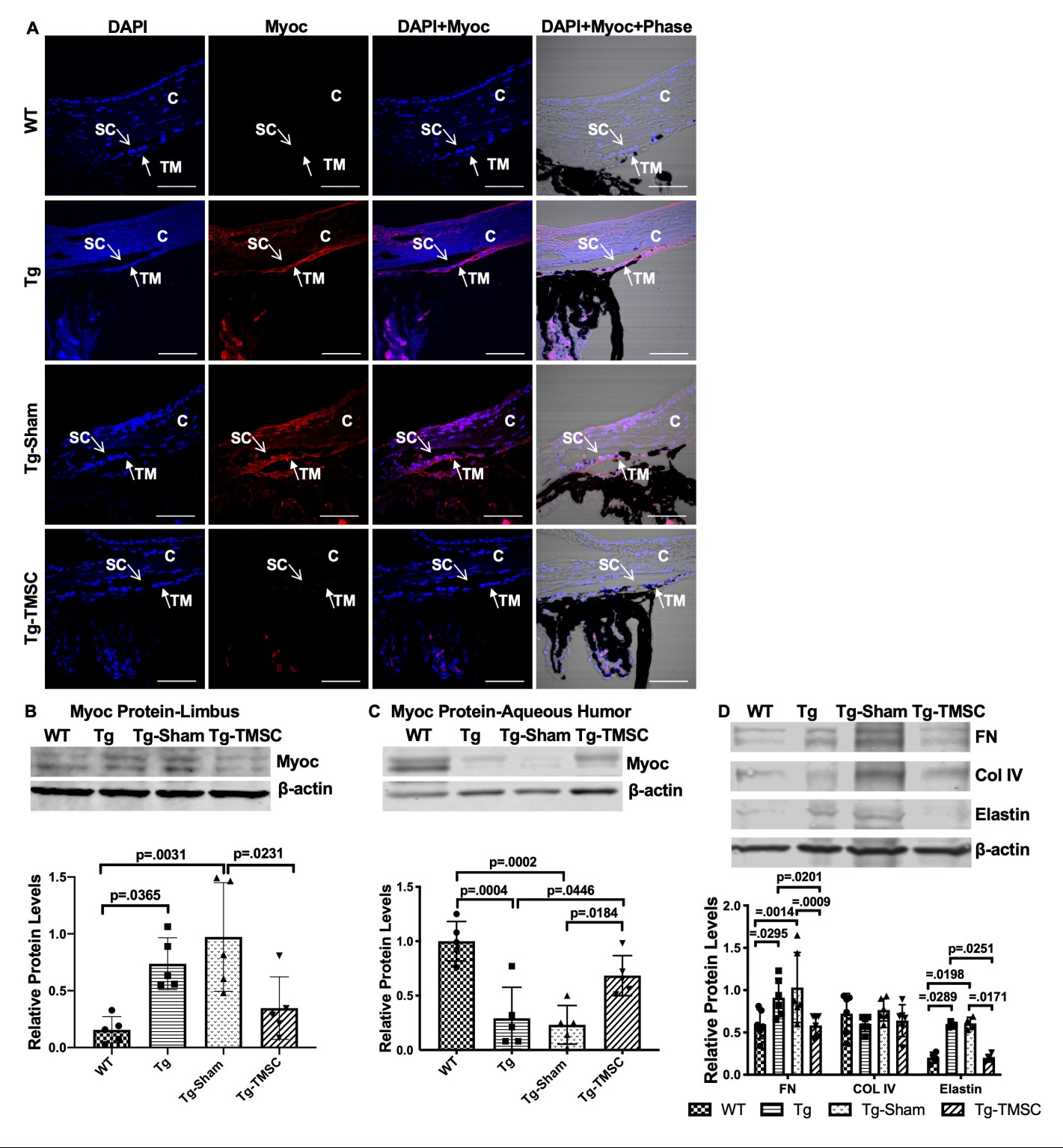

**Figure 5.** TMSCs reduce the Myoc retention in the TM tissue, promote the Myoc secretion into the aqueous humor, and reverse the ECM expression in the Tg-MyocY437H mice. (**A**) Immunofluorescent staining shows accumulated Myoc in the TM, iris, and ciliary body of the Tg and Tg-sham mice. TMSC transplantation alleviated the aggregation of Myoc in the TM, similar to the WT mice. Scale bars, 50 mm. Western blotting results show: (**B**) The representative bands of Myoc expression in the mouse limbal tissue and the relative Myoc protein levels with b-actin as internal control (n=5). (**C**) The representative bands of Myoc expression in the mouse aqueous humor and the relative Myoc protein levels with b-actin as internal control (n=5). (**D**) The representative bands of the expression of ECM components fibronectin (FN), collagen IV, and elastin in the limbal tissue and the relative ECM

*Figure 5 continued on next page*

*Figure 5 continued*

protein levels with b-actin as internal control (n=4-6). Data are presented as mean ± SD. One-way ANOVA (**B,C**) or two-way ANOVA (**D**) followed by Tukey's multiple comparisons test. C: cornea, SC: Schlemm's canal, TM: trabecular meshwork.

The online version of this article includes the following source data for figure 5:

**Source data 1.** Relative myocilin protein levels in the corneal limbus for *Figure 5B*.

We further evaluated the ultrastructure of the TM tissue by transmission electron microscopy (TEM) and measured the size of the ER and calculated as the ER area divided by the perimeter ($nm^2$/nm). As shown in *Figure 6B* (arrows) and calculation in *Figure 6C*, Tg-MyocY437H mice (24.53 ± 2.81 $nm^2$/nm) and Tg-Sham mice (24.50 ± 3.79 $nm^2$/nm) presented enlarged ER lumen as compared to the WT mice (7.03 ± 0.54 $nm^2$/nm). In contrast, the ER lumen of Tg-MyocY437H mice with TMSC transplantation was significantly reduced to 13.78 ± 1.02 $nm^2$/nm as compared to untreated (p=0.049) and Sham treated Tg-MyocY437H mice (p=0.0111) and more closely resembled that of WT mice (p=0.1423).

## TMSCs neither stimulate proliferation nor reverse ER stress of mutant TM cells in vitro

To further explore the mechanisms behind regenerative effect of transplanted TMSCs in vivo via increasing TM cellularity, enhancing Myoc secretion, remodeling the TM ECM and improving ER stress in the TM, we evaluated the effects of TMSCs on transduced MyocY437H mutant TM cells in vitro to detect the interactions between TMSCs and TM cells. TM cells were transduced with lentivirus which co-expressed GFP and Myoc with Y437H mutation (*Figure 7—figure supplement 1*). Transfected GFP-positive cells were then sorted using flow cytometry and further passaged as a

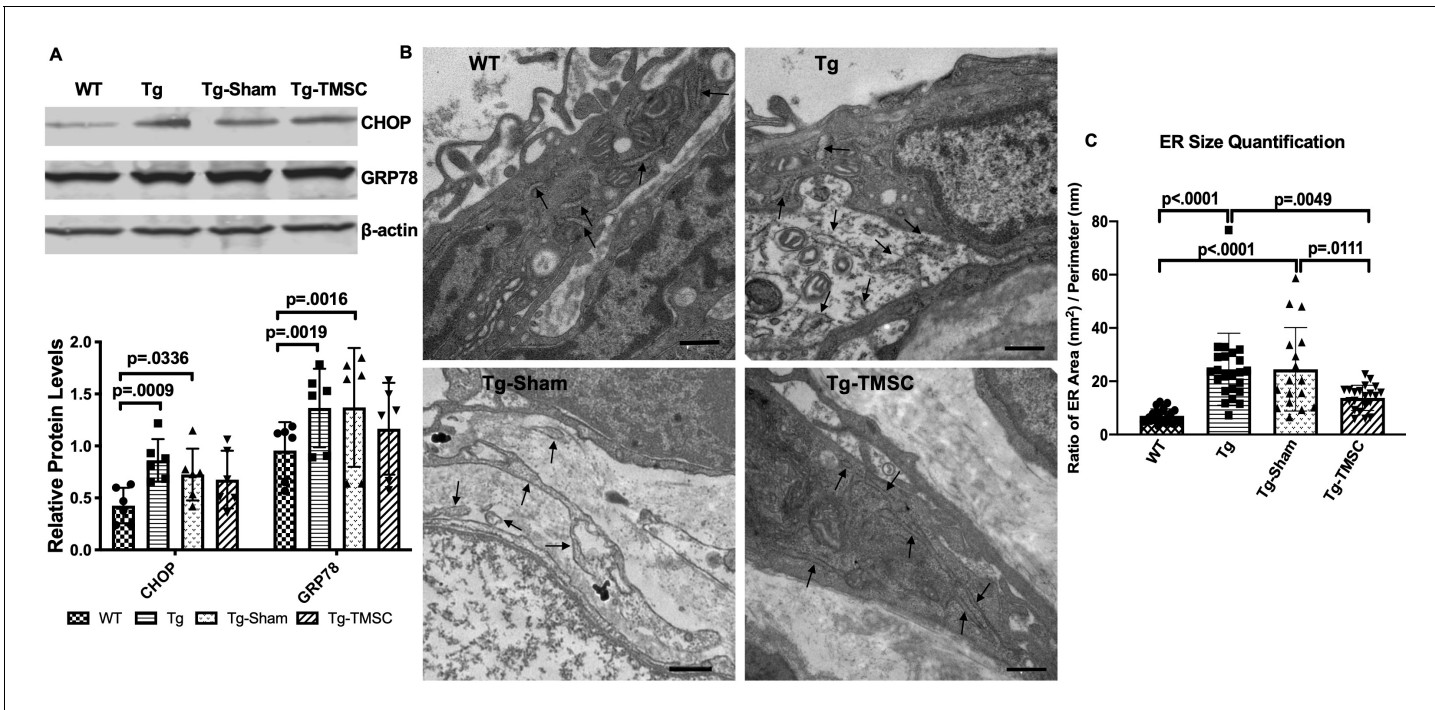

**Figure 6.** The effect of TMSCs on ER stress and ultrastructure of the TM in the Tg-Myoc Y437H mice. (**A**): Western blotting results show the representative bands of CHOP and GRP78 expression in the mouse limbal tissue and the relative protein levels with β-actin as internal control (n = 6). (**B**) TEM results indicates the ultrastructure of mouse TM tissue (40,000x) with black arrows pointing to the ER. Scale Bar = 500 nm. (**C**) ER size quantification calculated as area ($nm^2$)/perimeter (nm) (n = 18–23). Data are presented as mean ± SD. Two-way ANOVA (**A**) or one-way ANOVA (**C**) followed by Tukey's multiple comparisons test.

The online version of this article includes the following source data for figure 6:

**Source data 1.** Relative ER stress protein levels in the corneal limbus for *Figure 6A*; Relative ER sizes of the TM cells for *Figure 6C*.

predominantly GFP+ population of TM cells (*Figure 7A*) and strongly expressed Myoc (*Figure 7B*). Cultured TM cells were confirmed via Western blotting by their responsiveness to Dex with increased expression of Myoc (*Figure 7B*) after 100 nM Dex treatment for 5 days, one of the characteristics of TM cells (*Keller et al., 2018*). MyocY437H expressing GFP-positive TM cells were cultured alone, with TMSCs in the Transwell inserts (*Figure 7C*) or in direct contact with TMSCs and further assessed. Proliferation of the transduced TM cells was evaluated through analysis of incorporation rates of the EdU after 2 hr incubation. Mutant Myoc transduced TM cells showed 6.53 ± 1.19% EdU positivity, while 6.06 ± 1.78% (p=0.8267) and 6.51 ± 1.63% (p=0.9932) of cells were EdU positive when mutant TM cells were co-cultured with TMSCs in a Transwell insert or in direct contact with TMSCs, respectively. This indicated that neither co-culturing nor direct contact with TMSCs could stimulate proliferation of mutant TM cells (*Figure 7D*).

Next, we evaluated whether TMSCs could reduce ER stress in the Myoc mutant TM cells. As shown in *Figure 7E–F*, higher expression of Myoc and ER stress markers GRP78 and CHOP was detected in the mutant TM cells as compared to normal TM cells and TM cells transduced with GFP only. Co-culturing TM cells with TMSCs in the Transwell insert could mimic the interactions seen in vivo between homed TMSCs and TM cells. The co-culturing had little effect on reducing ER stress or promoting Myoc secretion in the mutant TM cells. The expression of GRP78, CHOP, and Myoc in the TM cells after co-culturing was similar to that without co-culturing (*Figure 7E–F*).

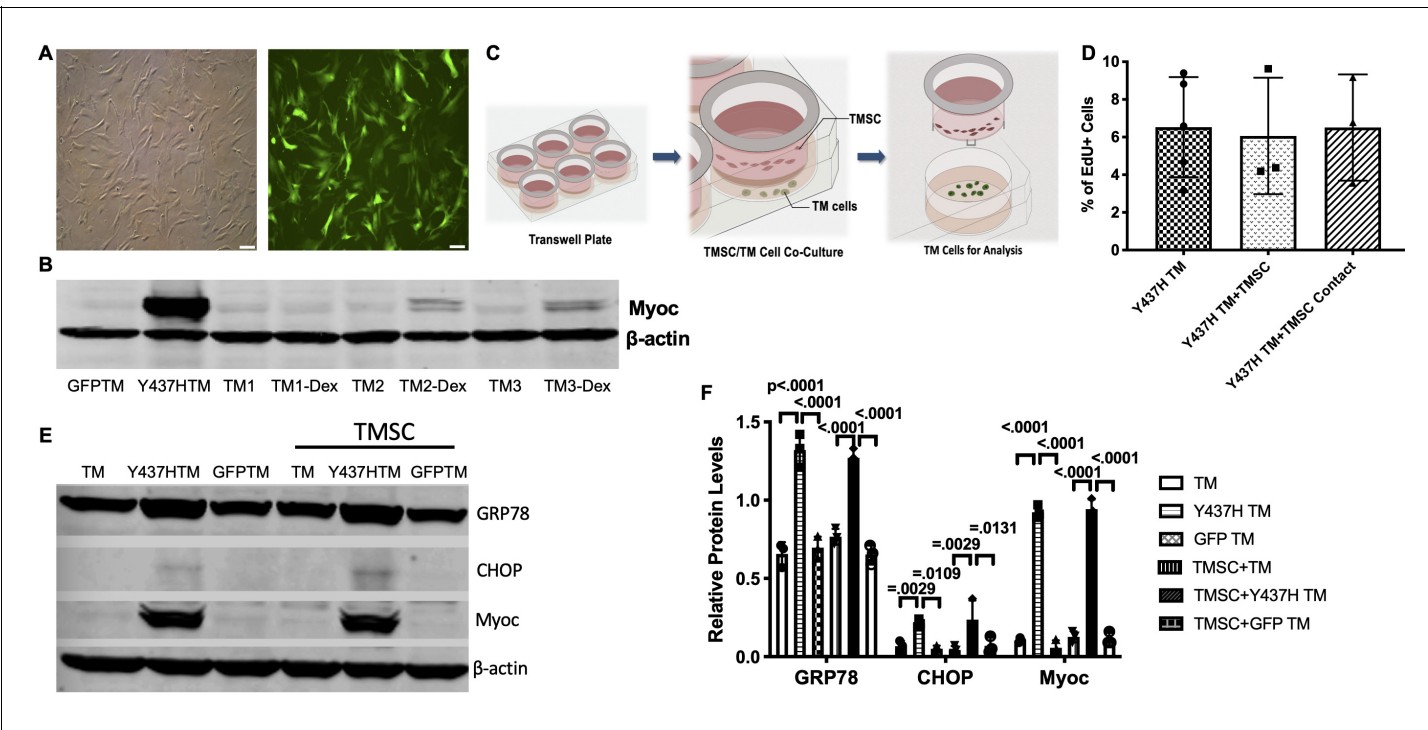

**Figure 7.** TMSCs could not reverse ER stress and stimulate proliferation of Myoc mutant TM cells in vitro. (A) The TM cells were transduced with recombinant lentivirus encoding GFP and Myoc Y437H mutation. The transduced GFP+ cells were sorted by Flow cytometry and the cultured sorted TM cells were almost 100% with GFP (green) in the cytoplasm. Scale Bars, 100 μm. (B) Transduced TM cells with Myoc Y437H mutation expressed high Myoc by western blotting and TM cells had increased Myoc expression after 5-day Dex treatment (TM2, TM3). TM1 did not have increased Myoc expression after Dex treatment so TM1 cells were discarded. (C) Schematic illustration shows co-culturing of TMSCs with TM cells for detection of TM cell changes. (D) Flow cytometry analysis of EdU incorporation shows neither co-culture nor direct contact with TMSCs for 4 days would affect TM cell proliferation (n=3-5). (E) Representative western blotting bands show the levels of ER stress markers and Myoc in the TM cells with or without TMSC co-culturing. (F) Relative protein levels with b-actin as internal control (n=3). Data are presented as mean ± SD. One-way ANOVA (D) or two-way ANOVA (F) followed by Tukey's multiple comparisons test.

The online version of this article includes the following source data and figure supplement(s) for figure 7:

**Source data 1.** Percentage of BrdU positive TM cells in different culture conditions for *Figure 7D*.
**Figure supplement 1.** Structure of lentiviral packaging plasmid: pLenti^CMV-Y437H-IRES-GFP.

## TMSCs differentiate into TM cells responsive to dexamethasone and gain phagocytic function under the ER stress condition in vitro

One of the mechanisms by which stem cells induce regeneration is differentiation into cells of desired lineage to compensate for deficient cells in the injured tissue. We previously reported that TMSCs (*Du et al., 2013*; *Xiong et al., 2020*; *Yun et al., 2018*) and ADSCs (*Zhou et al., 2020*) could differentiate into TM cells and express TM markers after homing to the TM tissues. However, the environment of the Tg-MyocY437H mouse TM where TMSCs stayed is different. The TM tissue of Tg-MyocY437H mice possesses ER stress with ECM changes (*Kasetti et al., 2016*). Understanding whether TMSCs can differentiate to TM cells under ER stress condition could help to elucidate how TMSCs regulate IOP in the Tg-MyocY437H mice. We co-cultured TMSCs together with the transduced Myoc mutant TM cells in a Transwell system (*Figure 8A*). CHI3L1 expression (*Figure 8B*) was significantly increased in the TMSCs after co-culturing with normal TM cells (TM+TMSC) or with MyocY437H mutant TM cells (Y437H TM+TMSC), in comparison to TMSCs without co-culture (TMSC). After another 7 day culture of the TMSCs in the presence of 100 nM Dex, the expression of Myoc (*Figure 8C*) was significantly increased in the TMSCs co-cultured with TM cells (TM+TMSC) or mutant TM cells (Y437H TM+TMSC) while the Myoc expression was almost undetectable without co-culture (TMSC). TMSCs co-cultured with normal and Myoc mutant TM cells gained the phagocytic function, evidenced by ingesting fluorescent labeled bioparticles (*Figure 8D–E*). Taken together, TMSCs are able to differentiate into TM cells responsive to dexamethasone treatment and possessing the phagocytic function under ER stress environment.

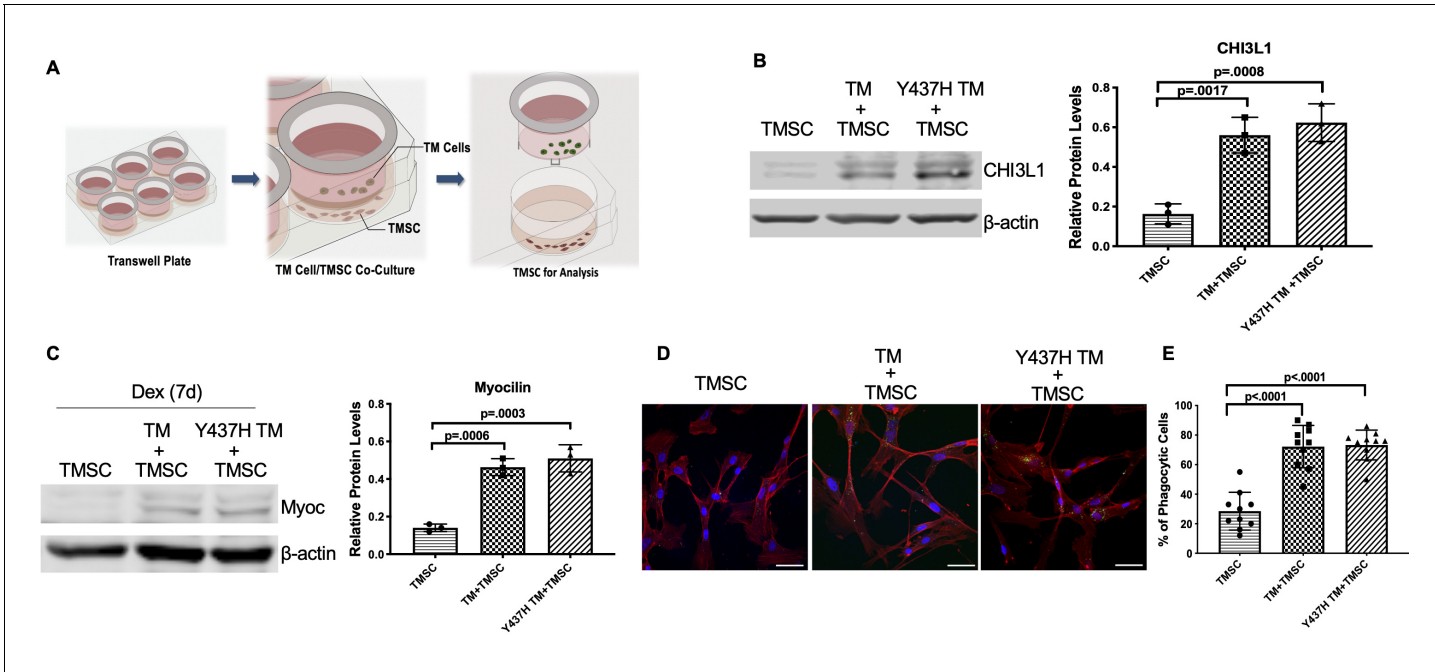

**Figure 8.** TMSCs differentiate into TM cells in vitro under ER stress environment. (**A**) Schematic illustration shows co-culturing of TMSCs with TM cells for detection of TMSC changes. (**B**) The expression of TM cell marker CHI3L1 was upregulated in the TMSCs after 10 days of co-culturing with normal TM cells or MyocY437H mutant TM cells (n = 3). (**C**) After co-culturing for 10 days, the co-cultured TM cells in the Transwell insert were removed, and TMSCs were further treated with Dex for another 7 days. The levels of Myoc were detected by western blotting and quantified (n = 3). (**D**) After co-culturing of TMSCs with the TM cells or Myoc Y437H mutant TM cells in the Transwell insert for 10 days, the phagocytic ability of the TMSCs was evaluated by ingestion of bioparticles shown green in the cytoplasm. Scale Bars, 50 μm. (**E**) Percentage of phagocytic cells averaged from 10 different views. Data are presented as mean ± SD. One-way ANOVA followed by Tukey's multiple comparisons test.

The online version of this article includes the following source data for figure 8:

**Source data 1.** Relative CHI3L1 protein levels by WB in TMSC with different culture conditions for *Figure 8B*.

# TMSCs had upregulated gene expression related to TM ECM maintenance and TM regeneration

We analyzed the transcriptomes of three individual TMSCs and fibroblasts from different donors since we previously reported that fibroblasts were not able to home to the TM and to regenerate the TM tissue (*Du et al., 2013*; *Yun et al., 2018*). We observed an upregulation of genes related to maintenance of TM ECM, integrity and motility like *integrin subunit alpha 3 (ITGA3), CHI3L1, vitronectin (VTN), lysyl oxidase (LOX), follistatin (FST), and collagen type IV alpha six chain (COL4A6)* (*Liton et al., 2006*) in TMSCs as compared to fibroblasts (*Figure 9A*). Top three upregulated pathways in TMSCs related to increased TM ECM interaction were (1) focal adhesion pathway including *VTN, collagen type IV alpha five chain (COL4A5), myosin light chain kinase (MYLK), platelet derived growth factor D (PDGFD), and COL4A6*; (2) PI3K-Akt signaling pathway including *VTN, COL4A5, PDGFD, and COL4A6*; and (3) ECM-receptor interaction pathway including *VTN, COL4A5, heparan sulfate proteoglycan core protein (HSPG2), and COL4A6* (*Supplementary file 1*). By interactome analysis for neuroprotective property of TMSCs, we identified many genes related to neuroprotection, including *neutralized E3 ubiquitin protein ligase 1 (NEURL1)* (formation of functional synapses), *neurofascin* (neurite extension, axonal guidance, synaptogenesis, myelination, and neuron-glial cell interactions), *neuroligin-1/3/4X* (synapse function and synaptic signal transmission). Reactome analysis identified proteins involved in glutamatergic, dopaminergic, GABAergic pathways activated in TMSCs (*Figure 9B*). Pathway enrichment analysis identified neurotrophin signaling pathway and PI3-Akt signaling pathway to be the major pathways related to the neuroprotection of RGCs.

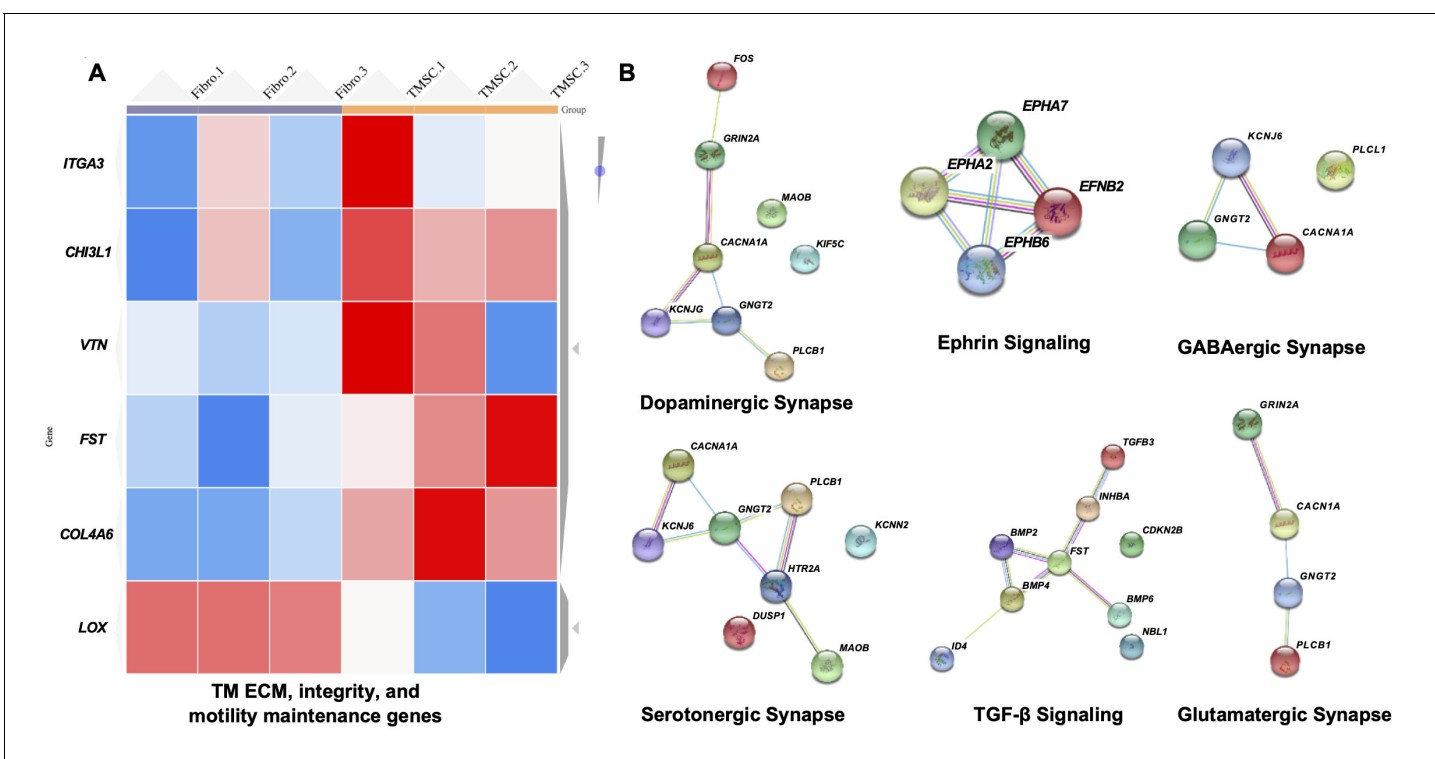

**Figure 9.** Transcriptome analysis of TM regeneration and neuroprotection genes among TMSC and fibroblasts. (**A**) Heatmap shows gene expression profile of TMSCs as compared to fibroblasts for genes involved in maintenance of TM extracellular matrix (ECM), TM integrity and motility, (false discover rate (FDR) < 1%, p<0.01). Scale, bright red squares: highest gene expression levels, bright blue squares: lowest gene expression levels. (**B**) Interactome analysis shows activation of different neuroprotection pathways in TMSCs as obtained by RNA sequencing analysis. Interactome networks were generated using STRING v11.

The online version of this article includes the following source data for figure 9:

**Source data 1.** Related gene expression increase (p<0.05) in three individual TMSCs as compared to fibroblasts for *Figure 9A*.

## Discussion

In this study we demonstrated, as illustrated in the graphical abstract *Figure 10*, that human TMSCs transplanted to the anterior chamber of transgenic Myoc Y437H mutant mice differentiated to the cells expressing TM cell markers at TM region and alleviated many of the parameters associated with glaucoma in the validated POAG mouse model. Specifically, transplantation of TMSCs reduced the IOP, increased the outflow facility, restored the RGC function with significantly improved pattern ERG and preserved the RGCs. With TMSC transplantation, the TM cellularity in the Tg-MyocY437H mice dramatically increased and the ECM components fibronectin and elastin dramatically reduced in comparison with untreated and sham injected Tg-MyocY437H mice. Although ER stress marker expression in the TM tissue was not significantly reduced after TMSC transplantation, the secreted Myoc into the aqueous humor was significantly increased, while non-secreted Myoc in the TM tissue was decreased to normal range compared to untreated and sham-treated Tg-MyocY437H mice. In vitro co-culturing study indicated that TMSCs could differentiate into Dex-responsive TM cells with phagocytic function in the presence of normal TM cells or transduced TM cells with Myoc Y437H mutation. The TMSCs did not reverse ER stress of cultured MyocY437H mutant TM cells in the co-culture platform. RNAseq analysis showing upregulation of genes related to TM regeneration including maintenance of TM integrity, motility, and ECM interaction in TMSCs as compared to fibroblasts might explain that TMSCs induce regeneration in the Tg-MyocY437H mice via modulation of ECM, promotion of TM integrity and motility, and increasing the oxidative stress defense mechanism of TM cells. In contrast, fibroblasts having much lower expression of the abovementioned genes were unable to produce any regenerative effect as we showed previously (*Yun et al., 2018*).

Our previous studies have shown that the response to ER stress inducers is different between TMSCs and TM cells (*Wang et al., 2019*), and TMSCs can differentiate into TM cells after homing in and retained in normal TM of WT mice (*Du et al., 2013*) and in the laser-damaged TM for regeneration (*Yun et al., 2018*). However, the microenvironment for the stem cells such as oxygen

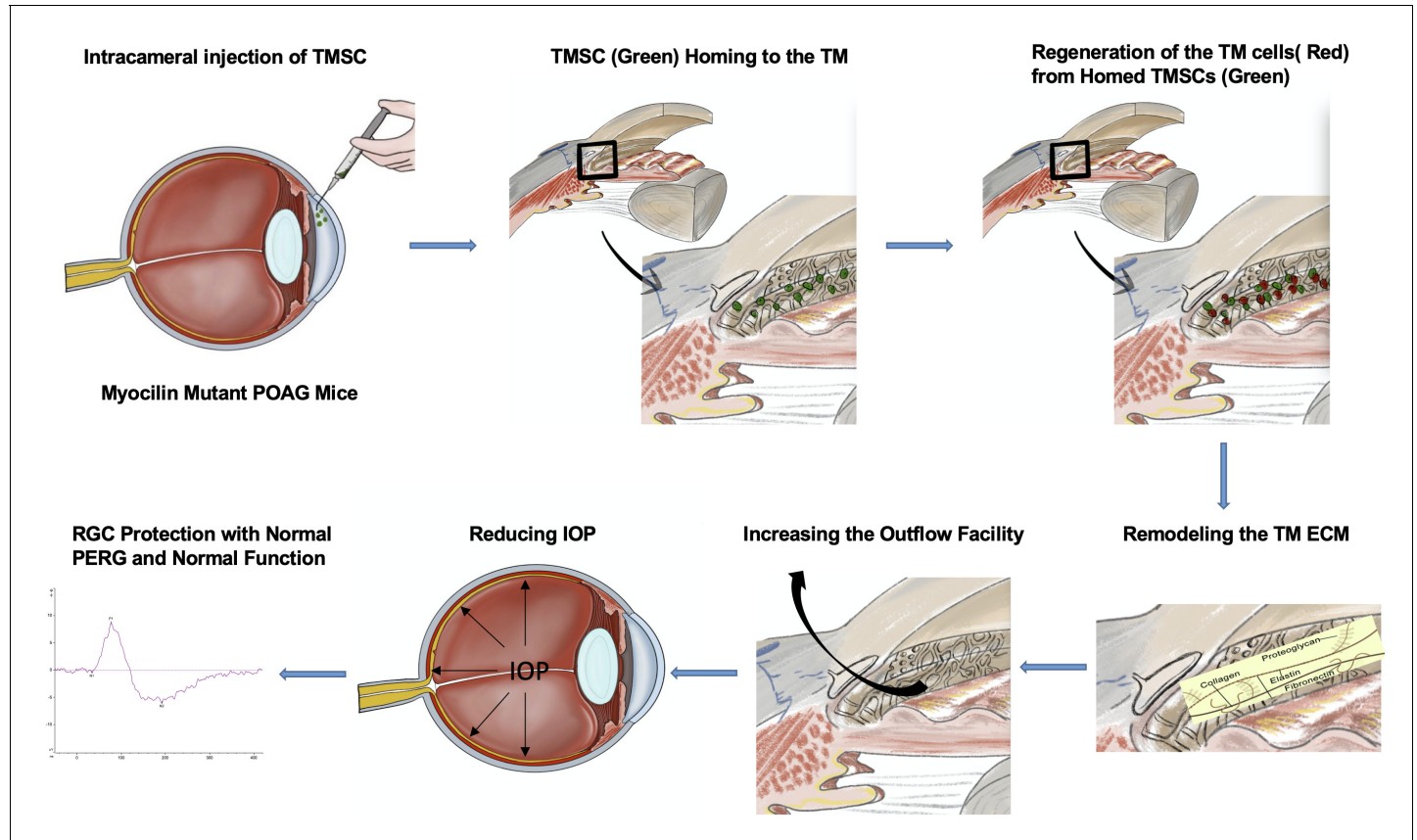

**Figure 10.** Graphical abstract.

concentration, PH value, osmotic pressure, proteases and cytokines, all affects the maintenance, survival, and regeneration properties of stem cells (*Urban, 2002*; *Wuertz et al., 2008*). It was therefore crucial to confirm whether TMSCs could convert to the TM cells in the Tg-MyocY437H mice with ER stress in the TM tissue. We demonstrated that TMSCs could differentiate into TM cells expressing the TM cell marker CHI3L1 (*Du et al., 2012*; *Kelley et al., 2009*; *Wang et al., 2019*) and possessed phagocytic ability 10 days after co-culturing with the transduced Myoc Y437H mutant TM cells. Phagocytic function of TM cells is responsible for ECM turnover and maintenance of the outflow pathway by removing cell debris, which is crucial for regulation of IOP. Moreover, the differentiated TMSCs were responsive to dexamethasone treatment with increased expression of Myoc, one of the important characteristics of TM cells (*Keller et al., 2018*). It indicates that TMSCs can successfully differentiate to functional TM cells under ER stress condition.

The homeostasis of the TM tissue is known to be important for maintaining IOP in the normal range (*Vranka et al., 2015*). Some pathological stimuli can elevate IOP by breaking down TM ECM homeostasis, which results in the excessive accumulation of ECM composition and insufficient degradation of ECM in the TM tissue, thereby decreasing the outflow facility (*Acott and Kelley, 2008*). ER stress arising from mutant Myoc aggregates in the ER can destroy assembly procedures of ECM proteins in the TM cells. The ECM components, such as fibronectin, elastin, and collagen IV were increased in the TM tissue due to abnormal ER function and pathological cellular status in the Tg-MyocY437H mice (*Kasetti et al., 2016*). Conversely, excessive ECM can aggravate ER stress in the TM cells, which can form a negative feedback to keep the chronic ER stress existing in the Tg-MyocY437H mice (*Kasetti et al., 2017*). We found that transplantation of TMSCs can reverse the expression of fibronectin and elastin to the normal levels in the Tg-MyocY437H mouse TM tissue. Although the reduction of ER stress marker expression after TMSC transplantation was not significant, the expression levels of Myoc reduced in the TM tissue and increased in the aqueous humor, and a large number of cells displayed normal ultrastructure without swollen ER in the TM region of the Tg-MyocY437H mice with TMSC transplantation. These observations suggest that differentiated healthy TM cells from TMSCs replaced the mutant TM cells and remodeled the ECM, improved the function of the diseased TM tissue, which increased the outflow facility and reduced IOP. A previous report (*Zhu et al., 2016*) indicated that transplantation of TM cells derived for iPSCs stimulated the endogenous TM cells to proliferate to increase the TM cellularity and reduce IOP. Although the cells and underlying mechanisms for the treatment in their study are different from ours, increased amount of TM cells was found in both studies. It indicates that restoration of TM cellularity and remodeling of the TM ECM are crucial for cell-based therapy for glaucoma.

Mesenchymal stem cells have been shown to reduce IOP in a laser-induced rat glaucoma model (*Manuguerra-Gagné et al., 2013*). The activation of progenitor cells in the ciliary body which can migrate and differentiate into TM cells in the damaged tissue might be induced by laser photocoagulation or injected cells. Further research is needed to elucidate whether TMSCs can recruit the endogenous stem cells to synergistically repair the TM tissue.

Loss of RGCs is responsible for the impairment of visual field and loss of visual acuity in POAG patients (*Rolle et al., 2014*; *Shoji et al., 2017*). Preserving the RGCs is as important as reducing IOP in treatment for glaucoma (*Sena and Lindsley, 2017*; *Stern et al., 2018*). It is also a critical parameter to evaluate whether stem cell-based therapy is suitable for the management of glaucoma (*Pearson and Martin, 2015*). Elevation of IOP and subsequent loss of RGCs were observed in the Tg-MyocY437H mice (*Zode et al., 2011*). Therefore, attention was also paid to the therapeutic effect of TMSCs on protection of RGC function. In this study, 90% of RGC function was saved 2 months after TMSC transplantation, while only 60% of RGC function remained in the 6-month-old Tg-MyocY437H mice without treatment as compared to age-matched WT mice. It indicates that TMSCs can prevent the RGCs from degeneration resulted from IOP elevation. Nevertheless, 10% loss of RGC function may be attributed to the delayed effect of TMSCs on reducing IOP in this glaucoma model, in which IOP starts to elevate from 3 months of age while TMSCs were transplanted at 4 months and the IOP reduction was observed at 5 months. Therefore, whether earlier intervention can achieve more RGC survival needs further investigation.

We previously reported that TMSCs can regenerate the damaged TM tissue while corneal fibroblasts did not repair the damaged TM tissue (*Yun et al., 2018*). Corneal fibroblasts did not express stem cell markers, such as NESTIN or OCT 4, which TMSCs were positive to and the fibroblasts did not home to and anchoring to the TM region after intracamerally injection (*Du et al., 2013*;

*Xiong et al., 2020*). All these suggest that TMSCs and corneal fibroblasts are distinctive in biological characteristics and behavior. Thus, we compared the transcriptomes between TMSCs and corneal fibroblasts.

Our transcriptome analysis indicates some ECM related genes and genes associated with ECM modulation pathways like PI3K-Akt signaling pathway, focal adhesion pathway, ECM-receptor interaction pathway (*Villegas et al., 2013*) were highly expressed in TMSCs. Whether TMSCs directly participate in ECM remodeling through aforementioned ECM related genes and pathways after transplantation remains to be elucidated. Since we detected that some TMSCs differentiated into TM cells after homing to the TM region, we speculate that both the differentiated TM cells from TMSCs and undifferentiated TMSCs participate in the TM ECM modulation. CHI3L1 has been involved in tissue remodeling and important for normal functioning of TM (*Kumar et al., 2020*; *Kumar et al., 2019*; *Yun et al., 2018*; *Zhou et al., 2020*). Integrins are crucial for ECM organization in the TM and help in anchoring of stem cells to the site of injury for regeneration (*Gagen et al., 2014*; *Xiong et al., 2020*). The upregulation of CHI3L1 and integrins in TMSCs as compared to fibroblasts further strengthen their regenerative role in the Tg-MyocY437H glaucoma model, although the CHI3L1 expression in TMSCs is much lower than that in TM cells.

Some of the genes which are upregulated in TMSCs have been shown to impart neuroprotective functions. CACNA1A uncovered in TMSC transcriptome (involved in dopaminergic, GABAergic, serotonergic and glutamatergic synapse) is responsible for communication between neurons by ion exchange and mutations in this gene results into neurological disorder (*Ophoff et al., 1996*). Similarly, KCNJ5 (GIRK2) are G-protein-gated potassium channels which are employed in control of hypothermia induced by activation of GABAergic, muscarinic, kappa opioid, adenosine, and serotonergic receptors (*Costa et al., 2005*). We speculate that the preservation of RGCs and their function in Tg-MyocY437H model by TMSCs is mainly due to reduced IOP, but these neuroprotective proteins through paracrine secretion by TMSCs might be also involved in the neuroprotective process. Further study is required to uncover it.

## Conclusion

Transplanted TMSCs can integrate into the TM tissue and differentiate into functional TM cells that can repopulate the TM tissue, remodel the TM ECM, and reinstate the TM homeostasis to restore the outflow facility, eventually reducing IOP and preserving RGCs and their function in the Tg-MyocY437H mouse model of POAG. Glaucoma with Myoc Y437H gene mutation as a subtype of POAG contains common pathophysiology of POAG that is reduced TM cellularity, which causes abnormal deposition of the ECM and increases IOP and damages the RGCs. Therefore, these results open an important avenue of a novel stem-cell-based strategy to eventually treat human open-angle glaucoma.

# Materials and methods

**Key resources table**

| Reagent type (species) or resource | Designation | Source or reference | Identifiers | Additional information |
|---|---|---|---|---|
| Genetic reagent (*Mus musculus*) | Transgenic Myoc Y437H mice | Courtesy of Dr. Gulab Zode *Zode et al., 2011* | | North Texas Eye Research Institute |
| Cell line (*Homo sapiens*) | Trabecular Meshwork Stem Cells (TMSCs) | This paper | | Cells isolated from both male and female donors, characterized, and maintained in Du lab |
| Cell line (*Homo sapiens*) | Trabecular Meshwork Cells (TM cells) | This paper | | |
| Cell line (*Homo sapiens*) | Corneal fibroblasts | This paper | | |

*Continued on next page*

*Continued*

| Reagent type (species) or resource | Designation | Source or reference | Identifiers | Additional information |
|---|---|---|---|---|
| Commercial assay or kit | Mycoplasma contamination detection kit | InvivoGen | Cat# rep-pt1 | |
| Antibody | Anti-Collagen IV (Rabbit polyclonal) | Sigma-Aldrich | Cat# SAB4500369 RRID:AB_10743858 | IF(1:100) WB(1:1000) |
| Antibody | Anti-AQP1 (Mouse monoclonal) | Santa Cruz Biotechnology | Cat# sc25287 RRID:AB_626694 | IF (1:100) |
| Antibody | Anti-human CHI3L1 (Goat polyclonal) | R and D Systems | Cat# AF2599, RRID:AB_2291883 | IF(1:50), WB (1:250) |
| Antibody | Anti-Ki67 (Rabbit polyclonal) | Abcam | Cat# ab15580 RRID:AB_443209 | IF(1:500) |
| Antibody | Anti-myocilin (Rabbit polyclonal) | Santa Cruz Biotechnology | Cat# Sc137233 RRID:AB_2148737 | IF(1:100) |
| Antibody | Anti-myocilin (Mouse monoclonal) | R and D Systems | Cat# MAB3446 RRID:AB_2148649 | WB(1:500) |
| Antibody | Anti-fibronectin (Rabbit polyclonal) | Abcam | Cat# b23750 RRID:AB_447655 | WB(1:1000) |
| Antibody | anti-elastin (Mouse monoclonal) | Millipore | Cat#: MAB2503 RRID:AB_2099602 | WB(1:500) |
| Antibody | CHOP (Mouse monoclonal) | Cell Signaling Technology | Cat#: 2895 RRID:AB_2089254 | WB(1:500) |
| Antibody | GRP78 (Mouse monoclonal) | Santa Cruz Biotechnology | Cat# sc-376768 RRID:AB_2819145 | WB(1:1000) |
| Antibody | β-actin (Mouse monoclonal) | Thermo Fisher | Cat# MA5-15739 RRID:AB_10979409 | WB(1:5000) |
| Recombinant DNA reagent | pLentiCMV-GFP (plasmid) | AddGene | Cat# 17448 | Lentiviral construct |
| Recombinant DNA reagent | pCAGIG2 (plasmid) | AddGene | Cat# 111159 | IRES-EGFP cassette |
| Recombinant DNA reagent | pcDNA3 Myoc Y437H | Courtesy of Dr. John Hulleman *Zadoo et al., 2016* | | UT Southwestern |
| Sequence-based reagent | Mouse DNA_F | Thermo Fisher | PCR primers | GACTAAGGCAAG AAAATGAGAATC |
| Sequence-based reagent | Mouse DNA _R | Thermo Fisher | PCR primers | CCTCTCCACTCC TGAGATAGC |
| Sequence-based reagent | Mutant Myoc_F | Thermo Fisher | PCR primers | ACAAAGGCAGGGTC GAGAAGACAGG |
| Sequence-based reagent | Mutant Myoc_R | Thermo Fisher | PCR primers | TTCCCACCTCTCTCT CCCCATGAGA |
| Commercial assay or kit | In Situ Cell Death Detection Kit | Sigma-Aldrich | Cat# 12156792910 | |
| Commercial assay or kit | RNeasy Mini Kit | Qiagen | Cat# 74106 | |
| Commercial assay or kit | cDNA Reverse Transcription Kit | Life Technologies | Cat# 4368813 | |

*Continued on next page*

*Continued*

| Reagent type (species) or resource | Designation | Source or reference | Identifiers | Additional information |
|---|---|---|---|---|
| Commercial assay or kit | Power SYBR Green PCR Master Mix | Life Technologies | Cat# 4368708 | |
| Chemical compound | opsonized Alexa 546-conjugated *S. aureus* bioparticles | Thermo Fisher | Cat# A10010 | |
| Software, algorithm | String V11 | https://string-db.org/ | RRID:SCR_005223 | |
| Software, algorithm | FlowJo version 10 | https://www.flowjo.com/ | RRID:SCR_008520 | |
| Software, algorithm | ImageJ | https://fiji.sc/ | RRID:SCR_002285 | |
| Software, algorithm | Graphpad Prism 8 | https://www.graphpad.com/scientific-software/prism/ | RRID:SCR_002798 | |
| Software, algorithm | Espion version 6 | http://diagnosysllc.com | | |
| Other | DAPI | Sigma-Aldrich | Cat# D9542 | Stain: 1 µg/ml |

## Animals

Four-month-old wildtype (WT) C57BL/6J mice were purchased from Jackson Laboratory as the normal control, and Tg-MyocY437H mice originated from C57BL/6J mice were kindly gifted by Dr. Gulab Zode (North Texas Eye Research Institute, Texas) and transferred to University of Pittsburgh. All the experiments conducted on the animals were approved by the University of Pittsburgh Institutional Animal Care and Use Committee (protocol 18022317) and complied with the ARVO Statement for the Use of Animals in Ophthalmic and Vision Research. Both WT C57B/6J and Tg-MyocY437H C57B/6J mice were bred in the animal facility at University of Pittsburgh. Mouse DNAs were isolated by biopsy from mouse ears for genotyping using the primers with sequences shown in the Key Resource Table (Mutant Myoc_F and _R; and Mouse DNA_F and _R). Mice with PCR product at 249 bp were regarded as carrying the Myoc mutation and a 610 bp product was used to confirm the content of mouse DNA (*Figure 1—figure supplement 1*).

## Intracameral transplantation of TMSCs and IOP measurement

Human TMSCs were isolated and passaged as previous reported (*Du et al., 2012*; *Wang et al., 2019*). Two TMSC strains from two different donors at passage three or four were used for cell injection. TMSCs were prelabeled with DiO at 50 µg/ml for 30 min (*Yun et al., 2018*) and thoroughly washed with DMEM/F12 and resuspended in the medium at the concentration of $1.67 \times 10^7$/ ml for injection. Mice were divided into four groups: Wildtype group (WT, n = 26), age-matched Tg-MyocY437H mice (Tg, n = 26), Tg mice with intracameral injection of the basal medium (Tg-Sham, n = 26) and Tg mice with TMSC transplantation (Tg-TMSC, n = 26). Intracameral injection was following previous published procedures (*Du et al., 2013*; *Yun et al., 2018*) with modifications. In brief, mice at the age of 4 months were anesthetized with ketamine-xylazine by intraperitoneal injection. A total of 3 µl of medium with $5 \times 10^4$ TMSCs or medium only (sham) were injected into the mouse anterior chamber using a 33-gauge needle connected to a 25 µl Hamilton syringe. An I-care tonometer was used to measure mouse IOP (TonoLab; Colonial Medical Supply, Windham, NH). Day-time IOP measurements were performed between 1:00 pm and 3:00 pm. Day-time IOP measurement before injection served as baseline and was conducted at different time points at week 1, week 2, month 1, and month 2 after transplantation. The night IOP was measured between 11 pm and 1 am and included two time points that were pre-transplantation as baseline and 2 months post-transplantation.

## Measurement of outflow facility

The procedure for measuring outflow facility was described previously (*Lei et al., 2011*; *Yun et al., 2018*; *Zhou et al., 2020*). All the outflow measurements on mouse eyes were finished within 6 hr after enucleation. Eyes were irrigated with phosphate buffer saline (PBS) at constant pressures of 4, 8, 15, and 25 mmHg and outflow was recorded at least 15 min at each pressure after the pressure was stable. Twelve eyes from each group were then perfused. Outflow facility (μL/min/mmHg) was calculated using the Goldmann equation (*Lei et al., 2011*). Data were accepted when $R^2$ was greater than 0.95 and data from at least six eyes per group were analyzed and averaged.

## Transmission electron microscopy

Transmission electron microscopy (TEM) was used to evaluate the ultrastructure of the TM as described previously (*Yun et al., 2014*). After removing the iris, the limbus tissues (n = 3) from each group were fixed in Karnovsky's fixative and divided into quarters of each tissue. Subsequently, the tissues were dehydrated and embedded in Epon and 65 nm Ultrathin sections were cut, stained with uranyl acetate (Electron Microscopy Sciences) and Reynold's lead citrate (Fisher). Sections were photographed at 80 kV on a Jeol 1011 TEM for analysis. For evaluation of ER size, the boundary of ER on each TEM image was delineated and ER region was colored by photoshop (Adobe). Then, the area and perimeter of the ER was calculated by Image pro plus (Media Cybernetics). The ER size was displayed as ER area/ER perimeter ($nm^2$/nm).

## Counting RGCs

The mouse eyes were enucleated and fixed in 4% paraformaldehyde overnight followed by subsequently dehydration and embedding in paraffin. A total of 5 μm sagittal sections were stained with hematoxylin and eosin. The sections adjacent to the optic nerve were used to capture the retina images using a ×40 oil objective in a microscopy (Olympus). The number of cells in the RGC layer was counted throughout the whole retina on four consecutive sections from each eye, and normalized to mean nuclei per mm.

## Immunostaining and counting of TM cellularity

The mouse eyes were fixed in 4% paraformaldehyde overnight and embedded in paraffin. After dewaxing, rehydration, heat-induced epitope retrieval and blocking with 10% heat-inactivated goat serum, sections were incubated with primary antibodies to myocilin, CHI3L1 (R and D Systems), collagen IV, Ki67(Abcam), AQP1(Santa Cruz) overnight at 4°C. After three washes with PBS, corresponding fluorescent secondary antibodies and 4′,6-diamidino-2-phenylindole (DAPI) were applied to the sections for 1 hr. After five washes, slides were mounted and imaged using a confocal microscope (Olympus IX81) and analyzed on FV10-ASW4.2 Viewer (Olympus). For measuring TM cellularity, primary antibody against collagen IV, together with phase-contrast images, were used to define the TM region in the sections. Cell nuclei stained with DAPI within the TM region were counted under FV10-ASW4.2 Viewer. Images of at least 10 fields per group were photographed, and the number of cell nuclei per field was counted and averaged.

## TUNEL analysis

The cell death detection kit (In Situ Cell Death Detection Kit, Sigma-Aldrich) was used to perform TUNEL analysis according to manufacturer's protocol. The cell nuclei on the section were stained with DAPI and images were captured under a confocal microscope (Olympus IX81). 10–14 sections from three eyes in each group were stained and analyzed for TMSC viability 2 months after transplantation.

## Phagocytosis assay

Cells were incubated with opsonized Alexa 546-conjugated *S. aureus* bioparticles (ThermoFisher) at a ratio of 20 bioparticles per cell at 37°C for 1 hr. After incubation, the cells were washed with PBS, trypsinized and transferred to another 6-well plated with coverslips at the bottom to get rid of any noningested bioparticles. After attachment, cells were fixed with 4% paraformaldehyde, permeabilized with 0.5% Triton X-100, and incubated with phalloidin conjugated with AlexFluor-647 and DAPI. Cellular phagocytosis of bioparticles which were ingested by the cells was observed within the

cytoplasm and photographed under a confocal microscope (Olympus). At least 10 individual views per condition were counted and averaged. The phagocytic ability was calculated as following:

$$\% \, of \, Phagocytic \, cells = \frac{Number \, of \, phagocytosed \, cells/field}{Total \, Cell \, Number/field} \times 100$$

## Construction of recombinant lentivirus Myoc Y437H/GFP

Recombinant Lentiviral Vector encoding Myoc Y437H was constructed from plasmids pLenti[CMV-GFP](Addgene 17448) (*Campeau et al., 2009*), p[CAGIG2] (Addgene 111159) (*Matsuda and Cepko, 2004*) and pcDNA3[Myoc Y437H] (*Zadoo et al., 2016*) (a kind gift from Dr. Hulleman at UT Southwestern). Briefly, pLenti [CMV-GFP-Puro] was digested with BamHI and SalI (NEB) to remove GFP cassette and served as the vector backbone, which was utilized to generate lentivirus encoding plasmid (pLenti[CMV-IRES-GFP]) by insertion of IRES-EGFP cassette (obtained from p[CAGIG2]) into it. The cDNA sequence containing Tyr437His mutation was amplified from pcDNA3[Myoc Y437H] by using the primers: Forward 5'-ACACCGACTCTAGAGATGAGGTTCTTCTGTGCACGT-3' and Reverse 5'-GGCGACCGGTGGATCTCACATCTTGGAGAGCTTGATG- 3'. It was subsequently cloned into BamHI site in the pLenti [CMV-IRES-GFP] by In-Fusion cloning kit (Clontech, 639649) to generate lentiviral packaging plasmids(pLenti[CMV-Y437H-IRES-GFP]), which were co-transfected with ViraPower[R] Lentiviral Packaging Mix (Invitrogen) in to 293 T cells using Lipofectamine 3000 Reagent (Invitrogen, L3000015) for Lentivirus assembly. The supernatant of transfected cells taken at day five was then concentrated with Lenti-X Concentrator (TakKaRa, 631232) for the collection of recombinant lentiviruses encoding the mutant myocilin (*Figure 7—figure supplement 1*).

## Cell culture and lentivirus transduction

Human cell culture was approved by the Committee for Oversignt of Research and Clinical Training Involving Decedents (CORID No. 161). The human corneas from both male and female donors containing TM tissue were obtained from the Center for Organ Recovery and Education (Pittsburgh, PA) and used for isolation of TM cells, TMSCs, and corneal fibroblasts. The cells were cultured and passaged as previously reported (*Du et al., 2009*; *Du et al., 2012*; *Wang et al., 2019*). Human TM cells were cultured in Dulbecco's modified Eagle's medium (DMEM)/F12 with 10% fetal bovine serum (FBS) and identified as responsiveness to 100 nM Dex treatment for 5 days with increased Myoc expression by Western blotting as recommended by a consensus of investigators in the field (*Keller et al., 2018*) and as we previously reported (*Xiong et al., 2020*). Human TMSCs were cultured in Opti-MEM (Invitrogen) with 5% FBS and a variety of supplements (*Du et al., 2012*) and identified as expression of stem cell markers CD73, CD90, CD105, CD166 by flow cytometry and confirmation of clonal formation and multipotency including the ability to differentiate into TM cells (*Du et al., 2012*; *Kumar et al., 2020*; *Yun et al., 2018*). Human fibroblasts were cultured in (DMEM)/F12 with 10% FBS and were negative to those stem cell markers without clonal formation (*Basu et al., 2014*; *Yun et al., 2018*). The TMSCs and TM cells at passages 3–4 and fibroblasts at passages 4–7 were used for the experiments in this study. All the cultured cells were tested negative for mycoplasma contamination using the Mycoplasma Detection Kit (Invivogen, San Diego, CA).

### Lentivirus transduction

Primary TM cells at passage 3% and 70% confluence were transduced with lentivirus encoding both mutant Myoc and GFP protein or GFP alone as a control at a multiplicity of infection (MOI) 3. Polybrene was used at 6 μg/ml to increase transduction efficiency. GFP positive cells were sorted through Flow cytometry (BD Biosciences, San Jose, CA) 3 days after transduction and passaged for the following studies.

### Co-culture of TMSCs and TM cells

For analyzing the effect of TMSCs on reversing ER stress and relieving accumulation of mutant Myoc in transduced TM cells, $5 \times 10^4$ TMSCs were seeded in the upper chamber of 6-well Corning Transwell inserts, while $5 \times 10^4$ normal TM cells or transduced TM cells with MyocY437H mutation were maintained at the bottom of the plates. To determine whether TMSCs could stimulate proliferation of mutant TM cells, $5 \times 10^4$ TMSCs were plated directly on pre-plated $5 \times 10^4$ Myoc mutant TM cells or TMSCs were in Transwell inserts as just described and cells were cocultured for 4 days. To

determine if TMSCs could differentiate into TM cells under ER stress environment, $5 \times 10^4$ mutant MyocY437H TM cells in the inserts were cocultured with $5 \times 10^4$ TMSCs for 10 days. Then, the upper inserts containing mutant TM cells were removed, the TMSCs in the bottom compartment were utilized for phagocytosis assay or further cultured with dexamethasone (100 nM) for another 7 days.

### EdU incorporation and flow cytometry analysis

To determine whether TMSCs influence the proliferation of mutant TM cells, the MyocY437H TM cells were cultured alone, with TMSCs in the Transwell inserts, or in direct contact with TMSCs. When cells reached 70% confluence, EdU was added into the culture medium to reach 10 µM concentration and incubated for 2 hr. The cells were then trypsinized, fixed with 4% paraformaldehyde, permeabilized with 0.5% of Triton X-100 and blocked with 1% bovine serum albumin (BSA). Subsequently, a cocktail containing sodium ascorbate (10 mM), azide-fluor 545 (8 µM), and copper sulfate (1 mM) was added and incubated for 10 min. Cells not undergoing the staining procedure, and cells incubating with azide-fluor 545 only were used as controls. Cell samples were run on the flow cytometer to gate both GFP+ mutant TM cells and EdU+ cells. The analysis was done using FlowJo_V10 software (FlowJo, Ashland, OR) and the percentage of EdU+ cells was counted as the number of GFP+EdU+ cells divided by GFP+ cells x 100. Each group was replicated at least three times.

### Western blotting analysis

Cultured cells, aqueous humor and mouse limbus tissue were lysed with RIPA buffer (Santa Cruz Biotechnology). BCA Protein Assay Kit (Pierce Biotechnology) was utilized for evaluating the concentration of proteins. A total of 30 µg total protein was loaded in each well and electrophoresed on the sodium dodecyl sulfate–polyacrylamide gel (ThermoFisher) and transferred to the PVDF membrane. After blocking in the blocking buffer, the membrane was incubated overnight with following primary antibodies accordingly, anti-CHI3L1, anti-Myoc, anti-elastin (Sigma), anti-Grp78, anti-fibronectin, and anti-collagen IV (Abcam). After washing with 0.1% Tween 20 in Tris-buffered saline for three times, it was incubated with secondary antibodies (IRDye 680LT and IRDye 800CW, LI-COR Biosciences). Fluorescent signals were captured on an infrared imager (Odyssey; LI-COR Biosciences). ImageJ was used for the densitometry analysis of protein expression. Each experiment was repeated three times.

### Anterior segment optical coherence tomography (OCT)

For evaluation of central corneal thickness and peripheral anterior synechia, an anterior segment optical coherence tomography (OCT; Visante OCT MODEL 1000; Carl Zeiss Meditec, Dublin, CA) was used. Eight eyes of each group were examined by determining quadrant-scans along four axes (0°–180°, 45°–225°, 90°–270°, and 135°–315°) to ensure scanning through the central cornea and data along the 0° to 180° axis were used for analysis.

### Pattern electroretinography (PERG)

PERG was performed on the Celeris apparatus (Diagnosys LLC, Lowell, MA) to evaluate the RGC function. Mice (n = 10 eyes for each group) were anesthetized with intraperitoneal injections of the mixture of ketamine and xylazine. The murine pupil was dilated with 0.5% tropicamide and 2.5% phenylephrine eye drops. A circular electrode centered on the cornea was placed in a plane perpendicular to the visual axis. Pattern stimuli consisted of horizontal bars of variable spatial frequencies and contrast that alternate at different temporal frequency. The parameters for PERG amplitude were spatial frequency 0.155 cycles/degree, temporal frequency 2.1 reversals/sec, contrast 100% and substantial averaging (600–1800 sweeps). The data were analyzed by the software Espion V6 (Diagnosys). The amplitude of P1 was used to analyze the function of RGCs.

### RNA sequencing

Three strains of cultured TMSCs and corneal fibroblasts isolated and cultured as previously described (*Du et al., 2009*; *Yun et al., 2018*) from different donors were lysed in RLT buffer (Qiagen). RNA isolation was performed using RNeasy mini kit (Qiagen) as per manufacturer's instructions. RNA pellet was treated with Ambion RNase-free DNase in DNase one buffer (Invitrogen). Final RNA pellet was dissolved in RNase-free diethyl pyrocarbonate (DEPC) water and sent to GENEWIZ,

LLC. (South Plainfield, NJ, USA) for RNA sequencing. The interactive heatmap was generated using Clustergrammer (*Fernandez et al., 2017*) which is freely available at http://amp.pharm.mssm.edu/clustergrammer/. Prior to displaying the heatmap, the raw gene counts were normalized using the logCPM method, filtered by selecting the genes with most variable expression, and finally transformed using the Z-score method with false discover rate (FDR) < 1%. Interactome networks were generated using STRING v11 (*Szklarczyk et al., 2019*).

## Statistical analysis

The results were expressed as mean ± standard deviation (SD). The statistical differences were analyzed by one-way or two-way ANOVA followed by Tukey's multiple comparisons test using Graph-Pad Prism 8. p<0.05 was considered statistically significance.

## Acknowledgements

The work was supported by NIH grants EY025643 (YD), P30-EY008098, Research to Prevent Blindness; and Eye and Ear Foundation (Pittsburgh, PA). The authors thank Kira Lathrop for assisting with confocal microscopy, Nancy Zurowski with Flow Cytometry, Katherine Davoli for paraffin sectioning, and Ming Sun for TEM sectioning.

## Additional information

### Competing interests

Yiqin Du: The University of Pittsburgh has a patent named "trabecular meshwork stem cells" with Yiqin Du as one of the inventors. The other authors declare that no competing interests exist.

### Funding

| Funder | Grant reference number | Author |
|---|---|---|
| National Eye Institute | EY025643 | Yiqin Du |
| National Eye Institute | P30-EY008098 | Yiqin Du |
| Research to Prevent Blindness | | Yiqin Du |
| Eye and Ear Foundation of Pittsburgh | | Yiqin Du |

The funders had no role in study design, data collection and interpretation, or the decision to submit the work for publication.

### Author contributions

Siqi Xiong, Conceptualization, Data curation, Formal analysis, Investigation, Methodology, Writing - original draft, Writing - review and editing; Ajay Kumar, Conceptualization, Formal analysis, Investigation, Writing - original draft, Writing - review and editing; Shenghe Tian, Validation, Investigation, Methodology, Writing - review and editing; Eman E Taher, Investigation, Writing - review and editing; Enzhi Yang, Investigation, Project administration, Writing - review and editing; Paul R Kinchington, Conceptualization, Data curation, Methodology, Writing - review and editing; Xiaobo Xia, Formal analysis, Validation, Writing - review and editing; Yiqin Du, Conceptualization, Resources, Data curation, Software, Formal analysis, Supervision, Funding acquisition, Validation, Investigation, Visualization, Methodology, Writing - original draft, Project administration, Writing - review and editing

### Author ORCIDs

Ajay Kumar https://orcid.org/0000-0003-3412-1823
Paul R Kinchington https://orcid.org/0000-0002-1901-9970
Yiqin Du https://orcid.org/0000-0002-4330-0957

## Ethics

Animal experimentation: All the experiments conducted on the animals were approved by the University of Pittsburgh Institutional Animal Care (protocol 18022317) and Use Committee and complied with the ARVO Statement for the Use of Animals in Ophthalmic and Vision Research. Human cell culture was approved by the Committee for Oversight of Research and Clinical Training Involving Decedents (CORID No. 161).

## Decision letter and Author response

Decision letter https://doi.org/10.7554/eLife.63677.sa1
Author response https://doi.org/10.7554/eLife.63677.sa2

## Additional files

### Supplementary files

• Supplementary file 1. Related gene expression increase (p<0.05) in genes related to TM ECM interaction. Related to *Figure 9*.

• Transparent reporting form

### Data availability

All data generated or analyzed during this study are included in the manuscript and supporting files. Source data files have been provided for Figures 1–9.

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
