## [Decision Letter]

**Acceptance summary:**

This study demonstrates the feasibility of stem cell therapy for glaucoma in an animal model.

**Decision letter after peer review:**

[Editors’ note: the authors submitted for reconsideration following the decision after peer review. What follows is the decision letter after the first round of review.]

Thank you for submitting your work entitled "Stem Cell Transplantation Rescued A Primary Open-Angle Glaucoma Mouse Model" for consideration by *eLife*. Your article has been reviewed by two peer reviewers, and the evaluation has been overseen by a Reviewing Editor and a Senior Editor. The reviewers have opted to remain anonymous.

Our decision has been reached after consultation between the reviewers. Based on these discussions and the individual reviews below, we regret to inform you that your work will not be considered further for publication in *eLife*.

Though both reviewers recognize the importance of this area of research, both have significant concerns regarding the quality of the data. In particular, the fate of the grafted cells, and the mechanism whereby the grafted cells achieve a therapeutic effect, has not been adequately addressed. It seems likely that revision to address these points and other key issues would likely entail very considerable experimental work.

Reviewer #2:

The authors report studies of human trabecular meshwork stem cells (TMSC) as a treatment for glaucoma in a transgenic mouse model of glaucoma caused by a myocilin mutation (Tyr437His). Tg-MyocY437H mice loose TM cells; developed reduced outflow facility and high IOP; and reduced PERG. The authors report that intracameral injection of TMSCs in these mice restores outflow facility, lowers IOP, and restores PERG. They also report that the TMSCs help to normalize secretion of Myoc. Normalization of extracellular matrix protein is also reported.

The authors report using stem cell therapy in a mouse model of glaucoma as a potential therapy for this disease, which is intriguing. Glaucoma is the most common cause of irreversible blindness and better therapies to protect vision are needed and this direction holds promise and would be of interest to the readership

My most important comments.

1) Stem cell-based therapeutic studies were previously reported in the same mouse model of glaucoma (Tg-Myoc-Tyr437His) by Zhu et al., 2016. The authors should compare and contrast their findings. The Zhu study reported increased numbers of endogenous TM cells, which raises the question of whether the TMSC delivered in the current study are directly responsible for a therapeutic effect or whether they similarly induce more endogenous TM cells. Are the increased cells seen in the TM in the current study from the TMSC or proliferation of endogenous TM cells? Are they even TM cells?

2) The authors should more clearly describe and characterize the cells that they are using (TMSC and primary TM cells) to authenticate their identity (i.e. pluripotency assays and markers for TM cells like steroid induction of MYOC)

3) The authors report PERG as a surrogate for functional damage of glaucoma in the Tg-Myoc-Tyr437His mice, but RGC counts and/or optic nerve axon counts is a more standard assessment for glaucoma in mice.

4) The analysis of the RNAseq data is inadequate. Significance should be discussed in the context of multiple measures or false discovery rate.

Reviewer #3:

In this paper, the authors use stem cell transplantation to rescue POAG. The premise is an exciting one for therapy. However, several shortcomings need to addressed before publication. Concerns and comments are listed below.

1) Please clarify mouse strains. Mouse strain C57BL/6J or N (not C57BL/6) makes a difference. What is the strain background of Tg-MyocY437H mice? Must compare within the same strain, preferably littermate given the variation of IOP, outflow, and PERG within the strain and between strains. Please report the exact p-values.

2) Day time IOP numbers between WT, Tg, and Tg-TMSC are modest at best. Tg IOP is elevated at night, according to the original paper describing the transgenic mouse. The authors should check the effect of TMSC implantation on IOP at night to claim reduction for the sake of completeness.

3) The authors should localize TMSC in the angle and demonstrate engraftment of the TM. If TMSCs are present in some other location, this should be shown as well. How can a reader know that the TMSC is even alive in the angle? Also, control should include injecting another unrelated stem cell type different from TMSC to show specificity.

4) In Figure 1D, why is the AC depth looking so different?

5) The outflow numbers are very high. The outflow number for C57Bl/6J mouse is around 3-5 nl/min/mmHg. Did the eyes dry out? Drying increases outflow during measurements.

6) For PERG, any reason why authors changed parameters from previously published numbers? Please report if there were any changes in latency.

7) In Figures 3 and 4, the sections are inferior quality. Space labeled SC is actually where the tissue has separated. The collagen IV staining does not look correct. ColIV is expected to mark the basement membrane. Why is it looking punctate? For looking at cellularity and tissue, the authors need to do a plastic section of eyes and grade the TM as in Libby (2003) 299-p1578. Hence the accuracy of the cell count is questionable. Are the position of the sections matched between the controls and the experimental eyes?

8) In Figure 4, the Myoc staining looks very different from the original 2011 Zode paper. WT Myco stain is barely visible. Please explain the discrepancy. Regarding the blot, there is a doublet in Myco blot. Which band was quantified? The same goes for panels D and F. Please show full gels for all blots, at least in supplement.

9) Please show higher magnification for ER in EM. For rigor, please quantify and report lumen size and percent of ER where lumen size changed. Again, the CHOP and GRP78 levels in the Tg animals are nowhere near as high as the Zode paper. Please explain.

10) A cartoon of co-culture technique will enhance understanding. Can the TMSC differentiate TM cells form CLANS? On this point, do the in vivo injected TMSC differentiate into TM cells? Do mouse TM cells induce human TMSC to differentiate into TM cells in vivo? (See point 3)

11) Please change "three strains of TMSC". All human TMSC. Please explain why you are sequencing fibroblasts and their origin. Please explain the inconsistency of the increase of specific genes between TM1, 2, and 3. A blanket statement that ECM genes are upregulated based on the data shown in Figure 8 is unwarranted.

12) The authors have not demonstrated the claim made in conclusion i.e., TMSC differentiates into TM cells in the mouse angle after injecting the stem cells, which is the premise on which the entire paper is geared.

[Editors’ note: further revisions were suggested prior to acceptance, as described below.]

Thank you for submitting your article "Stem Cell Transplantation Rescued A Primary Open-Angle Glaucoma Mouse Model" for consideration by *eLife*. Your article has been reviewed by one peer reviewer, and the evaluation has been overseen by a Reviewing Editor and Marianne Bronner as the Senior Editor. The reviewers have opted to remain anonymous.

The reviewers have discussed the reviews with one another and the Reviewing Editor has drafted this decision to help you prepare a revised submission.

Summary:

The study reports a preclinical model of cellular therapy for an important eye disorder, and reports possible mechanisms for the therapeutic effects observed

Revisions:

1) Figure 2: SC in the frozen sections is better but still does not match the quality of the image that is standard in the field. Specifically WT and Tg Sham. The other two are just passable.

2) Figure 5: same issue as in 1.

---

## [Author Response]

[Editors’ note: the authors resubmitted a revised version of the paper for consideration. What follows is the authors’ response to the first round of review.]

Reviewer #2:The authors report studies of human trabecular meshwork stem cells (TMSC) as a treatment for glaucoma in a transgenic mouse model of glaucoma caused by a myocilin mutation (Tyr437His). Tg-MyocY437H mice loose TM cells; developed reduced outflow facility and high IOP; and reduced PERG. The authors report that intracameral injection of TMSCs in these mice restores outflow facility, lowers IOP, and restores PERG. They also report that the TMSCs help to normalize secretion of Myoc. Normalization of extracellular matrix protein is also reported.The authors report using stem cell therapy in a mouse model of glaucoma as a potential therapy for this disease, which is intriguing. Glaucoma is the most common cause of irreversible blindness and better therapies to protect vision are needed and this direction holds promise and would be of interest to the readershipMy most important comments.1) Stem cell-based therapeutic studies were previously reported in the same mouse model of glaucoma (Tg-Myoc-Tyr437His) by Zhu et al., 2016. The authors should compare and contrast their findings. The Zhu study reported increased numbers of endogenous TM cells, which raises the question of whether the TMSC delivered in the current study are directly responsible for a therapeutic effect or whether they similarly induce more endogenous TM cells. Are the increased cells seen in the TM in the current study from the TMSC or proliferation of endogenous TM cells? Are they even TM cells?

Thanks for pointing this out. The Zhu study reported the injection of iPSC derived TM-like cells while we reported the injection of human TMSC. The cells are different. In our study, we showed that neither direct contact nor co-culture without contact of TMSC with TM cells could promote TM cells proliferation in vitro (Figure 7). In the co-culturing system, TMSC can differentiate to the cells expressing TM cell markers and responding to dexamethasone with increased myocilin expression; while TMSC cultured alone without TM cells didn't show any changes (Figure 8). In our new added Figure 4, it shows that many DiO-labeled TMSCs were integrated into the TM tissue, similar to our previous finding (Du, 2013, Yun 2018, Xiong 2020). Some of these DiO-labeled TMSCs expressed TM markers AQP1 and CHI3L1 (new added Figure 4). Our data indicate that after transplantation, the increased TM cellularity is mainly due to differentiated TM cells from TMSC, which remodeled extracellular matrix , rescued outflow facility and protected retinal ganglion cells through decreasing IOP. We have added the results in the revised manuscript and added a discussion as below:

“A previous report(Zhu et al., 2016) indicated that transplantation of TM cells derived for iPSCs stimulated the endogenous TM cell to proliferate to increase the TM cellularity and reduce IOP. Although the cells and underlying mechanisms for the treatment in their study are different from ours, increased amount of TM cells was found in both studies. It indicates that restoration of TM cellularity and remodeling of the TM ECM are crucial for cell-based therapy for glaucoma.”

2) The authors should more clearly describe and characterize the cells that they are using (TMSC and primary TM cells) to authenticate their identity (i.e. pluripotency assays and markers for TM cells like steroid induction of MYOC)

Thanks for pointing this out. We do characterization on all the TMSCs and TM cells we use. We added the following in the manuscript:

“Human TMSCs were isolated as previously described (Yun et al., 2018) and characterized by flow cytometry to confirm the positive expression of stem cell markers CD73, CD90, CD105, CD166, and negative expression of CD34 and CD45 as previously reported(Kumar et al., 2020; Yun et al., 2018).”.

We added a new figure as Figure 7B for the TM characterization and added the related method into the Materials and methods:

“Cultured TM cells were confirmed via Western blotting by their responsiveness to Dex with increased expression of Myoc (Figure 7B) after 100 nM Dex treatment for 5 days, one of the characteristics of TM cells(Keller et al., 2018).”.

3) The authors report PERG as a surrogate for functional damage of glaucoma in the Tg-Myoc-Tyr437His mice, but RGC counts and/or optic nerve axon counts is a more standard assessment for glaucoma in mice.

Thanks for the suggestion. We counted RGCs in the retina sections and put the data as Figure 2C. We added the description in the Materials and methods and added the results as:

“Furthermore, we counted the RGC numbers on 5-µm paraffin sections (Figure 2C, Figure 2—figure supplement 1). There were 70.48±2.26 RGC cells/mm in the retina of 6-month old WT mice, and 49.22±1.79 cells/mm in that of 6-month old Tg-MyocY437H mice with RGC loss (p<0.0001). The RGCs were preserved/rescued by TMSC transplantation in Tg-MyocY437H mice with the RGC number increased to 60.60±1.25 cells/mm (p<0.0001 as compared to untreated Tg-MyocY437H mice). This confirms that TMSC transplantation prevented/rescued the RGC loss and preserved the RGC function in the Tg-MyocY437H mice.”

4) The analysis of the RNAseq data is inadequate. Significance should be discussed in the context of multiple measures or false discovery rate.

Thanks for the advice. We reanalyzed the RNAseq data as new Figure 9 and edited the Discussion as:

“We previously reported that TMSCs can regenerate the damaged TM tissue while corneal fibroblasts did not repair the damaged TM tissue(Yun et al., 2018). […] We speculate that the preservation of RGCs and their function in Tg-MYOCY437H model by TMSCs is mainly due to reduced IOP, but these neuroprotective proteins through paracrine secretion by TMSCs might be also involved in the neuroprotective process. Further study is required to uncover it.”

Reviewer #3:In this paper, the authors use stem cell transplantation to rescue POAG. The premise is an exciting one for therapy. However, several shortcomings need to addressed before publication. Concerns and comments are listed below.1) Please clarify mouse strains. Mouse strain C57BL/6J or N (not C57BL/6) makes a difference. What is the strain background of Tg-MyocY437H mice? Must compare within the same strain, preferably littermate given the variation of IOP, outflow, and PERG within the strain and between strains. Please report the exact p-values.

Both the WT and Tg-MyocY437H mice were C57BL/6J mice. We bred the

Tg-MyocY437H mice with WT C57BL/6J mice in house so every mouse is the same strain. We have added this description in the Materials and methods:

“Four-month old wildtype (WT) C57BL/6J mice were purchased from Jackson Laboratory as the normal control, and Tg-MyocY437H mice originated from C57BL/6J mice were kindly gifted by Dr. Gulab Zode (North Texas Eye Research Institute, Texas) and transferred to University of Pittsburgh. All the experiments conducted on the animals were approved by the University of Pittsburgh Institutional Animal Care and Use Committee and complied with the ARVO Statement for the Use of Animals in Ophthalmic and Vision Research. Both WT C57B/6J and Tg-MyocY437H C57B/6J mice were bred in the animal facility at University of Pittsburgh. We have reported the exact p-values in each related figure.”

2) Day time IOP numbers between WT, Tg, and Tg-TMSC are modest at best. Tg IOP is elevated at night, according to the original paper describing the transgenic mouse. The authors should check the effect of TMSC implantation on IOP at night to claim reduction for the sake of completeness.

We measured night IOP before the treatment and 60 days after TMSCs transplantation.

Data were presented as Figure 1B and the results were added to the text:

“Meanwhile, we measured mouse night IOP which was more obviously elevated than day-time IOP as reported before (Zode et al., 2011). In consistent with the day-time IOP, there was a significant difference of the baseline night IOP between 4-month old Tg-MyocY437H mice (Figure 1B, 17.73±2.25 mmHg) and age-matched WT mice (13.67±2.77 mmHg, P<0.0001). 2 months after TMSC transplantation, the night IOP of Tg-MyocY437H mice (11.75±2.83 mmHg) reduced to the same level as that of WT mice (11.55±2.52 mmHg, P=0.9925).”

3) The authors should localize TMSC in the angle and demonstrate engraftment of the TM. If TMSCs are present in some other location, this should be shown as well. How can a reader know that the TMSC is even alive in the angle? Also, control should include injecting another unrelated stem cell type different from TMSC to show specificity.

In the revised manuscript, we showed that the injected DiO labeled TMSCs were still detectable and alive in TM region by TUNEL staining (Figure 4C) with expression of TM cell markers AQP1 and CHI3L1 (Figure 4A).

We have previously reported that intracamerally injected fibroblasts were not specifically homed to the TM region and they had gone apoptosis after transplantation (Du 2013, Yun 2018). We added the following in the Discussion:

“We previously reported that TMSCs can regenerate the damaged TM tissue while corneal fibroblasts did not repair the damaged TM tissue(Yun et al., 2018). Corneal fibroblasts did not express stem cell markers, such as NESTIN or OCT 4, which TMSCs were positive to and the fibroblasts did not home to and anchoring to the TM region after intracamerally injection (Du et al., 2013; Xiong et al., 2020). All these suggest that TMSCs and corneal fibroblasts are distinctive in biological characteristics and behavior.”

In Discussion, we have added the following:

“RNAseq analysis showing upregulation of genes related to TM regeneration including maintenance of TM integrity, motility, and ECM interaction in TMSCs as compared to fibroblasts might explain that TMSCs induce regeneration in the Tg-MyocY437H mice via modulation of ECM, promotion of TM integrity and motility, and increasing the oxidative stress defense mechanism of TM cells. In contrast, fibroblasts having much lower expression of the abovementioned genes were unable to produce any regenerative effect as we showed previously(Yun et al., 2018).”

4) In Figure 1D, why is the AC depth looking so different?

We didn't find the difference of AC depth among the groups in the OCT pictures averaged of all the eyes measured. The OCT analysis was utilized to exclude possibility of IOP elevation due to angle closure arising from TMSC transplantation or due to the central corneal depth changes. We have changed it to a more representative figure.

5) The outflow numbers are very high. The outflow number for C57Bl/6J mouse is around 3-5 nl/min/mmHg. Did the eyes dry out? Drying increases outflow during measurements.

We were very careful to keep the eyes moisture throughout the whole perfusion and outflow facility measurement process. Outflow result in this study is consistent with our previous published studies(Zhou et al., 2020 and Yun et al, 2018)

6) For PERG, any reason why authors changed parameters from previously published numbers? Please report if there were any changes in latency.

We did PERG on the Celeris apparatus from the Diagnosys LLC and all the parameters analyzed were from the software Epsion V6 from the same company. We added this description in Materials and methods. We didn’t find any latency difference between groups.

7) In Figures 3 and 4, the sections are inferior quality. Space labeled SC is actually where the tissue has separated. The collagen IV staining does not look correct. ColIV is expected to mark the basement membrane. Why is it looking punctate? For looking at cellularity and tissue, the authors need to do a plastic section of eyes and grade the TM as in Libby (2003) 299-p1578. Hence the accuracy of the cell count is questionable. Are the position of the sections matched between the controls and the experimental eyes?

We admit that plastic sections are the best for counting TM cell numbers. We sectioned the eyes with the same direction and counted the TM cells in the sections when optic nerve could be seen so the counting was in the similar position. In some of the sections, the SC space widened because of tissue processing. With the Col IV staining and the bright field images, the TM region cannot be messed up with any adjacent structures. Collagen IV stains both the basement membrane and extracellular matrix in the TM as a fibrotic marker (Vranka et al., 2015). We have changed the figures which have better quality. In Zhu et al., 2016 paper cryosection was used for immunostaining analysis including counting TM cells number. According to the published data and our experience, we believe our TM cellularity counting on the paraffin sections was accurate and plastic section is not mandatory for the TM cellularity analysis.

8) In Figure 4, the Myoc staining looks very different from the original 2011 Zode paper. WT Myco stain is barely visible. Please explain the discrepancy. Regarding the blot, there is a doublet in Myco blot. Which band was quantified? The same goes for panels D and F. Please show full gels for all blots, at least in supplement.

In current study, Myoc staining was very weak at the TM region in the WT mice and staining intensity increased in Tg group, which was similar to the 2011 Zode paper. According to the published article(N Jacobson , M Andrews, A R Shepard et al. Non-secretion of mutant proteins of the glaucoma gene myocilin in cultured trabecular meshwork cells and in aqueous humor. Hum Mol Genet. 2001, 15;10:117-25), intracellular Myoc protein manifest as two band 55 and 57 kDa, secreted myocilin is variable seen as two bands of 53 and 55KD or single band of 53-55 kDa. We calculated the band density including both bands. Full gels for western blot were submitted for review.

9) Please show higher magnification for ER in EM. For rigor, please quantify and report lumen size and percent of ER where lumen size changed. Again, the CHOP and GRP78 levels in the Tg animals are nowhere near as high as the Zode paper. Please explain.

Thanks for the suggestion. For the ER size, please refer to the answer to reviewer 2’s question 2.

In Zode paper, the CHOP and GRP78 levels didn't normalized to internal control. According to gel picture in his paper, the difference of GRP78 protein level between WT and Tg group was not as obvious as CHOP level, which is similar to our result. We speculate the time of explosion while capturing picture or the picture contrast gives rise to different gel looking between ours and Zode’s.

10) A cartoon of co-culture technique will enhance understanding. Can the TMSC differentiate TM cells form CLANS? On this point, do the in vivo injected TMSC differentiate into TM cells? Do mouse TM cells induce human TMSC to differentiate into TM cells in vivo? (See point 3)

We added cartoon to explain the co-culture technique (Figure 7C and Figure 8A). After co-culturing with myocilin mutant TM cells, TMSCs expressed TM cell markers and possessed more potent phagocytic ability, and myocilin was increased in the TMSCs after dexamethasone treatment for. In contrast, TMSCs without co-culturing didn't show aforementioned changes. We didn't examine CLAN formation in TMSC differentiated TM cells. According to the suggestion, we also confirmed that transplanted TMSCs can express AQP1 and CHI3L1 at the TM tissue two month after injection, indicating TMSCs can differentiate into TM cells in Tg mice. We can speculate mouse TM cells can stimulate human TMSC to differentiate into TM cells in vivo according to the results from in vitro study.

11) Please change "three strains of TMSC". All human TMSC. Please explain why you are sequencing fibroblasts and their origin. Please explain the inconsistency of the increase of specific genes between TM1, 2, and 3. A blanket statement that ECM genes are upregulated based on the data shown in Figure 8 is unwarranted.

We did RNAseq on 3 strains of TMSCs and 3 strains of fibroblasts.

The fibroblast used in this study was originated from donor human cornea. We changed “fibroblasts” as “corneal fibroblasts” in the manuscript and added the description in the Materials and methods:

“Three strains of cultured TMSCs and corneal fibroblasts isolated and cultured as previously described (Du et al., 2009; Yun et al., 2018) from different donors were lysed in RLT buffer (Qiagen).”

We added the description in the text:

“We previously reported that TMSCs can regenerate the damaged TM tissue while corneal fibroblasts did not repair the damaged TM tissue(Yun et al., 2018). Corneal fibroblasts did not express stem cell markers, such as NESTIN or OCT 4, which TMSCs were positive to and the fibroblasts did not home to and anchoring to the TM region after intracamerally injection (Du et al., 2013; Xiong et al., 2020). All these suggest that TMSCs and corneal fibroblasts are distinctive in biological characteristics and behavior. Thus we compared the transcriptomes between TMSCs and corneal fibroblasts.”

The RNAseq data were from different TMSCs and fibroblasts from different donors so there are variations among donors. We re-analyzed the RNAseq data and changed the Results and the Discussion.

12) The authors have not demonstrated the claim made in conclusion i.e., TMSC differentiates into TM cells in the mouse angle after injecting the stem cells, which is the premise on which the entire paper is geared.

Thanks for the suggestion. In the revision, we have added a new Figure 4 to show the TMSCs differentiates into TM cells expressing AQP1 and CHI3L1. Please refer to the answer to the reviewer 2’s question 1.

We have reworded the conclusions in the manuscript to make it clear:

“Conclusion

Transplanted TMSCs can integrate into the TM tissue and differentiate into functional TM cells that can repopulate the TM tissue, remodel the TM ECM, and restore the TM homeostasis to resolve the outflow facility, eventually reducing IOP and preserving RGCs and their function in the Tg-MyocY437H mouse model of POAG. Myoc mutation glaucoma as a subtype of POAG contains common pathophysiology of POAG that is reduced TM cellularity, which causes abnormal deposition of the ECM and increases IOP and damages the RGCs. Therefore, these results open an important avenue of a novel stem cell-based strategy to eventually treat human open-angle glaucoma.”

[Editors’ note: what follows is the authors’ response to the second round of review.]

Revisions:1) Figure 2: SC in the frozen sections is better but still does not match the quality of the image that is standard in the field. Specifically WT and Tg Sham. The other two are just passable.

Thanks for pointing this out. Since Figure 2 does not contain any staining results, I believe the reviewer meant the figures in Figure 3. We have replaced the staining figures of WT and Tg Sham in Figure 3 and marked the Schlemm’s canal clearer with arrows.

2) Figure 5: same issue as in 1.

We have replaced the staining figures of WT and Tg Sham in Figure 5 and marked the Schlemm’s canal clearer with arrows.